# Online Multiclass Boosting

**Young Hun Jung**　　　　　　　**Jack Goetz**　　　　　　　**Ambuj Tewari**
Department of Statistics
University of Michigan
Ann Arbor, MI 48109
{yhjung, jrgoetz, tewaria}@umich.edu

## Abstract

Recent work has extended the theoretical analysis of boosting algorithms to multi-class problems and to online settings. However, the multiclass extension is in the batch setting and the online extensions only consider binary classification. We fill this gap in the literature by defining, and justifying, a weak learning condition for online multiclass boosting. This condition leads to an optimal boosting algorithm that requires the minimal number of weak learners to achieve a certain accuracy. Additionally, we propose an adaptive algorithm which is near optimal and enjoys an excellent performance on real data due to its adaptive property.

## 1　Introduction

*Boosting* methods are a ensemble learning methods that aggregate several (not necessarily) weak learners to build a stronger learner. When used to aggregate reasonably strong learners, boosting has been shown to produce results competitive with other state-of-the-art methods (e.g., Korytkowski et al. [1], Zhang and Wang [2]). Until recently theoretical development in this area has been focused on batch binary settings where the learner can observe the entire training set at once, and the labels are restricted to be binary (cf. Schapire and Freund [3]). In the past few years, progress has been made to extend the theory and algorithms to more general settings.

Dealing with *multiclass classification* turned out to be more subtle than initially expected. Mukherjee and Schapire [4] unify several different proposals made earlier in the literature and provide a general framework for multiclass boosting. They state their weak learning conditions in terms of *cost matrices* that have to satisfy certain restrictions: for example, labeling with the ground truth should have less cost than labeling with some other labels. A weak learning condition, just like the binary condition, states that the performance of a learner, now judged using a cost matrix, should be better than a random guessing baseline. One particular condition they call the *edge-over-random* condition, proves to be sufficient for boostability. The edge-over-random condition will also figure prominently in this paper. They also consider a necessary and sufficient condition for boostability but it turns out to be computationally intractable to be used in practice.

A recent trend in modern machine learning is to train learners in an *online setting* where the instances come sequentially and the learner has to make predictions instantly. Oza [5] initially proposed an online boosting algorithm that has accuracy comparable with the batch version, but it took several years to design an algorithm with theoretical justification (Chen et al. [6]). Beygelzimer et al. [7] achieved a breakthrough by proposing an optimal algorithm in online binary settings and an adaptive algorithm that works quite well in practice. These theories in online binary boosting have led to several extensions. For example, Chen et al. [8] combine one vs all method with binary boosting algorithms to tackle online multiclass problems with bandit feedback, and Hu et al. [9] build a theory of boosting in regression setting.

In this paper, we combine the insights and techniques of Mukherjee and Schapire [4] and Beygelzimer et al. [7] to provide a framework for online multiclass boosting. The cost matrix framework from the former work is adopted to propose an online weak learning condition that defines how well a learner can perform over a random guess (Definition 1). We show this condition is naturally derived from its batch setting counterpart. From this weak learning condition, a boosting algorithm (Algorithm 1) is proposed which is theoretically optimal in that it requires the minimal number of learners and sample complexity to attain a specified level of accuracy. We also develop an adaptive algorithm (Algorithm 2) which allows learners to have variable strengths. This algorithm is theoretically less efficient than the optimal one, but the experimental results show that it is quite comparable and sometimes even better due to its adaptive property. Both algorithms not only possess theoretical proofs of mistake bounds, but also demonstrate superior performance over preexisting methods.

## 2 Preliminaries

We first describe the basic setup for online boosting. While in the batch setting, an additional weak learner is trained at every iteration, in the online setting, the algorithm starts with a fixed count of $N$ *weak learners* and a *booster* which manages the weak learners. There are $k$ possible labels $[k] := \{1, \cdots, k\}$ and $k$ is known to the learners. At each iteration $t = 1, \cdots, T$, an *adversary* picks a labeled example $(\mathbf{x}_t, y_t) \in \mathcal{X} \times [k]$, where $\mathcal{X}$ is some domain, and reveals $\mathbf{x}_t$ to the booster. Once the booster observes the unlabeled data $\mathbf{x}_t$, it gathers the weak learners' predictions and makes a final prediction. Throughout this paper, index $i$ takes values from 1 to $N$; $t$ from 1 to $T$; and $l$ from 1 to $k$.

We utilize the *cost matrix framework*, first proposed by Mukherjee and Schapire [4], to develop multiclass boosting algorithms. This is a key ingredient in the multiclass extension as it enables different penalization for each pair of correct label and prediction, and we further develop this framework to suit the online setting. The booster sequentially computes *cost matrices* $\{\mathbf{C}_t^i \in \mathbb{R}^{k \times k} \mid i = 1, \cdots, N\}$, sends $(\mathbf{x}_t, \mathbf{C}_t^i)$ to the $i^{th}$ weak learner $WL^i$, and gets its prediction $l_t^i \in [k]$. Here the cost matrix $\mathbf{C}_t^i$ plays a role of loss function in that $WL^i$ tries to minimize the cumulative cost $\sum_t \mathbf{C}_t^i[y_t, l_t^i]$. As the booster wants each learner to predict the correct label, it wants to set the diagonal entries of $\mathbf{C}_t^i$ to be minimal among its row. At this stage, the true label $y_t$ is not revealed yet, but the previous weak learners' predictions can affect the computation of the cost matrix for the next learner. Given a matrix $\mathbf{C}$, the $(i, j)^{th}$ entry will be denoted by $\mathbf{C}[i, j]$, and $i^{th}$ row vector by $\mathbf{C}[i]$.

Once all the learners make predictions, the booster makes the final prediction $\hat{y}_t$ by majority votes. The booster can either take simple majority votes or weighted ones. In fact for the adaptive algorithm, we will allow weighted votes so that the booster can assign more weights on well-performing learners. The weight for $WL^i$ at iteration $t$ will be denoted by $\alpha_t^i$. After observing the booster's final decision, the adversary reveals the true label $y_t$, and the booster suffers 0-1 loss $\mathbb{1}(\hat{y}_t \neq y_t)$. The booster also shares the true label to the weak learners so that they can train on this data point.

Two main issues have to be resolved to design a good boosting algorithm. First, we need to design the booster's strategy for producing cost matrices. Second, we need to quantify weak learner's ability to reduce the cumulative cost $\sum_{t=1}^T \mathbf{C}_t^i[y_t, l_t^i]$. The first issue will be resolved by introducing potential functions, which will be thoroughly discussed in Section 3.1. For the second issue, we introduce our online weak learning condition, a generalization of the weak learning assumption in Beygelzimer et al. [7], stating that for any adaptively given sequence of cost matrices, weak learners can produce predictions whose cumulative cost is less than that incurred by random guessing. The online weak learning condition will be discussed in the following section. For the analysis of the adaptive algorithm, we use empirical edges instead of the online weak learning condition.

### 2.1 Online weak learning condition

In this section, we propose an online weak learning condition that states the weak learners are better than a random guess. We first define a baseline condition that is better than a random guess. Let $\Delta[k]$ denote a family of distributions over $[k]$ and $\mathbf{u}_\gamma^l \in \Delta[k]$ be a uniform distribution that puts $\gamma$ more weight on the label $l$. For example, $\mathbf{u}_\gamma^1 = (\frac{1-\gamma}{k} + \gamma, \frac{1-\gamma}{k}, \cdots, \frac{1-\gamma}{k})$. For a given sequence of examples $\{(\mathbf{x}_t, y_t) \mid t = 1, \cdots, T\}$, $\mathbf{U}_\gamma \in \mathbb{R}^{T \times k}$ consists of rows $\mathbf{u}_\gamma^{y_t}$. Then we restrict the booster's

choice of cost matrices to

$$\mathcal{C}_1^{eor} := \{\mathbf{C} \in \mathbb{R}^{k \times k} \mid \forall l, r \in [k], \ \mathbf{C}[l, l] = 0, \mathbf{C}[l, r] \geq 0, \ \text{and} \ ||\mathbf{C}[l]||_1 = 1\}.$$

Note that diagonal entries are minimal among the row, and $\mathcal{C}_1^{eor}$ also has a normalization constraint. A broader choice of cost matrices is allowed if one can assign importance weights on observations, which is possible for various learners. Even if the learner does not take the importance weight as an input, we can achieve a similar effect by sending to the learner an instance with probability that is proportional to its weight. Interested readers can refer Beygelzimer et al. [7, Lemma 1]. From now on, we will assume that our weak learners can take weight $w_t$ as an input.

We are ready to present our online weak learning condition. This condition is in fact naturally derived from the batch setting counterpart that is well studied by Mukherjee and Schapire [4]. The link is thoroughly discussed in Appendix A. For the scaling issue, we assume the weights $w_t$ lie in $[0, 1]$.

**Definition 1. (Online multiclass weak learning condition)** *For parameters $\gamma, \delta \in (0, 1)$, and $S > 0$, a pair of online learner and an adversary is said to satisfy online weak learning condition with parameters $\delta, \gamma$, and $S$ if for any sample length $T$, any adaptive sequence of labeled examples, and for any adaptively chosen series of pairs of weight and cost matrix $\{(w_t, \mathbf{C}_t) \in [0, 1] \times \mathcal{C}_1^{eor} \mid t = 1, \cdots, T\}$, the learner can generate predictions $\hat{y}_t$ such that with probability at least $1 - \delta$,*

$$\sum_{t=1}^{T} w_t \mathbf{C}_t[y_t, \hat{y}_t] \leq \mathbf{C} \bullet \mathbf{U}'_\gamma + S = \frac{1 - \gamma}{k} ||\boldsymbol{w}||_1 + S, \tag{1}$$

*where $\mathbf{C} \in \mathbb{R}^{T \times k}$ consists of rows of $w_t \mathbf{C}_t[y_t]$ and $\mathbf{A} \bullet \mathbf{B}'$ denotes the Frobenius inner product $Tr(\mathbf{A}\mathbf{B}')$. $\boldsymbol{w} = (w_1, \cdots, w_T)$ and the last equality holds due to the normalized condition on $\mathcal{C}_1^{eor}$. $\gamma$ is called an edge, and $S$ an excess loss.*

**Remark.** *Notice that this condition is imposed on a pair of learner and adversary instead of solely on a learner. This is because no learner can satisfy this condition if the adversary draws samples in a completely adaptive manner. The probabilistic statement is necessary because many online algorithms' predictions are not deterministic. The excess loss requirement is needed since an online learner cannot produce meaningful predictions before observing a sufficient number of examples.*

## 3 An optimal algorithm

In this section, we describe the booster's optimal strategy for designing cost matrices. We first introduce a general theory without specifying the loss, and later investigate the asymptotic behavior of cumulative loss suffered by our algorithm under the specific 0-1 loss. We adopt the potential function framework from Mukherjee and Schapire [4] and extend it to the online setting. Potential functions help both in designing cost matrices and in proving the mistake bound of the algorithm.

### 3.1 A general online multiclass boost-by-majority (OnlineMBBM) algorithm

We will keep track of the weighted cumulative votes of the first $i$ weak learners for the sample $\mathbf{x}_t$ by $\mathbf{s}_t^i := \sum_{j=1}^{i} \alpha_t^j \mathbf{e}_{l_t^j}$, where $\alpha_t^i$ is the weight of $WL^i$, $l_t^i$ is its prediction and $\mathbf{e}_j$ is the $j^{th}$ standard basis vector. For the optimal algorithm, we assume that $\alpha_t^i = 1, \ \forall i, t$. In other words, the booster makes the final decision by simple majority votes. Given a cumulative vote $\mathbf{s} \in \mathbb{R}^k$, suppose we have a loss function $L^r(\mathbf{s})$ where $r$ denotes the correct label. We call a loss function *proper*, if it is a decreasing function of $\mathbf{s}[r]$ and an increasing function of other coordinates (we alert the reader that "proper loss" has at least one other meaning in the literature). From now on, we will assume that our loss function is proper. A good example of proper loss is multiclass 0-1 loss:

$$L^r(\mathbf{s}) := \mathbb{1}(\max_{l \neq r} \mathbf{s}[l] \geq \mathbf{s}[r]). \tag{2}$$

The purpose of the potential function $\phi_i^r(\mathbf{s})$ is to estimate the booster's loss when there remain $i$ learners until the final decision and the current cumulative vote is $\mathbf{s}$. More precisely, we want potential functions to satisfy the following conditions:

$$\begin{aligned}
\phi_0^r(\mathbf{s}) &= L^r(\mathbf{s}), \\
\phi_{i+1}^r(\mathbf{s}) &= \mathbb{E}_{l \sim \mathbf{u}_\gamma^r} \phi_i^r(\mathbf{s} + \mathbf{e}_l).
\end{aligned} \tag{3}$$

---

**Algorithm 1** Online Multiclass Boost-by-Majority (OnlineMBBM)

---

1: **for** $t = 1, \cdots, T$ **do**
2:     Receive example $\mathbf{x}_t$
3:     Set $\mathbf{s}_t^0 = \mathbf{0} \in \mathbb{R}^k$
4:     **for** $i = 1, \cdots, N$ **do**
5:         Set the normalized cost matrix $\mathbf{D}_t^i$ according to (5) and pass it to $WL^i$
6:         Get weak predictions $l_t^i = WL^i(\mathbf{x}_t)$ and update $\mathbf{s}_t^i = \mathbf{s}_t^{i-1} + \mathbf{e}_{l_t^i}$
7:     **end for**
8:     Predict $\hat{y}_t := \mathrm{argmax}_l\, \mathbf{s}_t^N[l]$ and receive true label $y_t$
9:     **for** $i = 1, \cdots, N$ **do**
10:        Set $\mathbf{w}^i[t] = \sum_{l=1}^k [\phi_{N-i}^{y_t}(\mathbf{s}_t^{i-1} + \mathbf{e}_l) - \phi_{N-i}^{y_t}(\mathbf{s}_t^{i-1} + \mathbf{e}_{y_t})]$
11:        Pass training example with weight $(\mathbf{x}_t, y_t, \mathbf{w}^i[t])$ to $WL^i$
12:     **end for**
13: **end for**

---

Readers should note that $\phi_i^r(\mathbf{s})$ also inherits the proper property of the loss function, which can be shown by induction. The condition (3) can be loosened by replacing both equalities by inequalities "$\geq$", but in practice we usually use equalities.

Now we describe the booster's strategy for designing cost matrices. After observing $\mathbf{x}_t$, the booster sequentially sets a cost matrix $\mathbf{C}_t^i$ for $WL^i$, gets the weak learner's prediction $l_t^i$ and uses this in the computation of the next cost matrix $\mathbf{C}_t^{i+1}$. Ultimately, booster wants to set

$$\mathbf{C}_t^i[r, l] = \phi_{N-i}^r(\mathbf{s}_t^{i-1} + \mathbf{e}_l). \tag{4}$$

However, this cost matrix does not satisfy the condition of $\mathcal{C}_1^{eor}$, and thus should be modified in order to utilize the weak learning condition. First to make the cost for the true label equal to 0, we subtract $\mathbf{C}_t^i[r, r]$ from every element of $\mathbf{C}_t^i[r]$. Since the potential function is proper, our new cost matrix still has non-negative elements after the subtraction. We then normalize the row so that each row has $\ell_1$ norm equal to 1. In other words, we get new normalized cost matrix

$$\mathbf{D}_t^i[r, l] = \frac{\phi_{N-i}^r(\mathbf{s}_t^{i-1} + \mathbf{e}_l) - \phi_{N-i}^r(\mathbf{s}_t^{i-1} + \mathbf{e}_r)}{\mathbf{w}^i[t]}, \tag{5}$$

where $\mathbf{w}^i[t] := \sum_{l=1}^k \phi_{N-i}^r(\mathbf{s}_t^{i-1} + \mathbf{e}_l) - \phi_{N-i}^r(\mathbf{s}_t^{i-1} + \mathbf{e}_r)$ plays the role of weight. It is still possible that a row vector $\mathbf{C}_t^i[r]$ is a zero vector so that normalization is impossible. In this case, we just leave it as a zero vector. Our weak learning condition (1) still works with cost matrices some of whose row vectors are zeros because however the learner predicts, it incurs no cost.

After defining cost matrices, the rest of the algorithm is straightforward except we have to estimate $||\mathbf{w}^i||_\infty$ to normalize the weight. This is necessary because the weak learning condition assumes the weights lying in $[0, 1]$. We cannot compute the exact value of $||\mathbf{w}^i||_\infty$ until the last instance is revealed, which is fine as we need this value only in proving the mistake bound. The estimate $w^{i*}$ for $||\mathbf{w}^i||_\infty$ requires to specify the loss, and we postpone the technical parts to Appendix B.2. Interested readers may directly refer Lemma 10 before proceeding. Once the learners generate predictions after observing cost matrices, the final decision is made by simple majority votes. After the true label is revealed, the booster updates the weight and sends the labeled instance with weight to the weak learners. The pseudocode for the entire algorithm is depicted in Algorithm 1. The algorithm is named after Beygelzimer et al. [7, OnlineBBM], which is in fact OnlineMBBM with binary labels.

We present our first main result regarding the mistake bound of general OnlineMBBM. The proof appears in Appendix B.1 where the main idea is adopted from Beygelzimer et al. [7, Lemma 3].

**Theorem 2. (Cumulative loss bound for OnlineMBBM)** *Suppose weak learners and an adversary satisfy the online weak learning condition (1) with parameters $\delta, \gamma$, and $S$. For any $T$ and $N$ satisfying $\delta \ll \frac{1}{N}$, and any adaptive sequence of labeled examples generated by the adversary, the final loss suffered by OnlineMBBM satisfies the following inequality with probability $1 - N\delta$:*

$$\sum_{t=1}^T L^{y_t}(\mathbf{s}_t^N) \leq \phi_N^1(\boldsymbol{0})T + S\sum_{i=1}^N w^{i*}. \tag{6}$$

Here $\phi_N^1(\mathbf{0})$ plays a role of asymptotic error rate and the second term determines the sample complexity. We will investigate the behavior of those terms under the 0-1 loss in the following section.

## 3.2 Mistake bound under 0-1 loss and its optimality

From now on, we will specify the loss to be multiclass 0-1 loss defined in (2), which might be the most relevant measure in multiclass problems. To present a specific mistake bound, two terms in the RHS of (6) should be bounded. This requires an approximation of potentials, which is technical and postponed to Appendix B.2. Lemma 9 and 10 provide the bounds for those terms. We also mention another bound for the weight in the remark after Lemma 10 so that one can use whichever tighter. Combining the above lemmas with Theorem 2 gives the following corollary. The additional constraint on $\gamma$ comes from Lemma 10.

**Corollary 3. (0-1 loss bound of OnlineMBBM)** *Suppose weak learners and an adversary satisfy the online weak learning condition (1) with parameters $\delta, \gamma$, and $S$, where $\gamma < \frac{1}{2}$. For any $T$ and $N$ satisfying $\delta \ll \frac{1}{N}$ and any adaptive sequence of labeled examples generated by the adversary, OnlineMBBM can generate predictions $\hat{y}_t$ that satisfy the following inequality with probability $1 - N\delta$:*

$$\sum_{t=1}^{T} \mathbb{1}(y_t \neq \hat{y}_t) \leq (k-1)e^{-\frac{\gamma^2 N}{2}}T + \tilde{O}(k^{5/2}\sqrt{N}S). \tag{7}$$

*Therefore in order to achieve error rate $\epsilon$, it suffices to use $N = \Theta(\frac{1}{\gamma^2} \ln \frac{k}{\epsilon})$ weak learners, which gives an excess loss bound of $\tilde{\Theta}(\frac{k^{5/2}}{\gamma}S)$.*

**Remark.** *Note that the above excess loss bound gives a sample complexity bound of $\tilde{\Theta}(\frac{k^{5/2}}{\epsilon\gamma}S)$. If we use alternative weight bound to get $kNS$ as an upper bound for the second term in (6), we end up having $\tilde{O}(kNS)$. This will give an excess loss bound of $\tilde{\Theta}(\frac{k}{\gamma^2}S)$.*

We now provide lower bounds on the number of learners and sample complexity for arbitrary online boosting algorithms to evaluate the optimality of OnlineMBBM under 0-1 loss. In particular, we construct weak learners that satisfy the online weak learning condition (1) and have almost matching asymptotic error rate and excess loss compared to those of OnlineMBBM as in (7). Indeed we can prove that the number of learners and sample complexity of OnlineMBBM is optimal up to logarithmic factors, ignoring the influence of the number of classes $k$. Our bounds are possibly suboptimal up to polynomial factors in $k$, and the problem to fill the gap remains open. The detailed proof and a discussion of the gap can be found in Appendix B.3. Our lower bound is a multiclass version of Beygelzimer et al. [7, Theorem 3].

**Theorem 4. (Lower bounds for $N$ and $T$)** *For any $\gamma \in (0, \frac{1}{4})$, $\delta, \epsilon \in (0, 1)$, and $S \geq \frac{k \ln(\frac{1}{\delta})}{\gamma}$, there exists an adversary with a family of learners satisfying the online weak learning condition (1) with parameters $\delta, \gamma$, and $S$, such that to achieve asymptotic error rate $\epsilon$, an online boosting algorithm requires at least $\Omega(\frac{1}{k^2\gamma^2} \ln \frac{1}{\epsilon})$ learners and a sample complexity of $\Omega(\frac{k}{\epsilon\gamma}S)$.*

## 4 An adaptive algorithm

The online weak learning condition imposes minimal assumptions on the asymptotic accuracy of learners, and obviously it leads to a solid theory of online boosting. However, it has two main practical limitations. The first is the difficulty of estimating the edge $\gamma$. Given a learner and an adversary, it is by no means a simple task to find the maximum edge that satisfies (1). The second issue is that different learners may have different edges. Some learners may in fact be quite strong with significant edges, while others are just slightly better than a random guess. In this case, OnlineMBBM has to pick the minimum edge as it assumes common $\gamma$ for all weak learners. It is obviously inefficient in that the booster underestimates the strong learners' accuracy.

Our adaptive algorithm will discard the online weak learning condition to provide a more practical method. Empirical edges $\gamma_1, \cdots, \gamma_N$ (see Section 4.2 for the definition) are measured for the weak learners and are used to bound the number of mistakes made by the boosting algorithm.

## 4.1 Choice of loss function

Adaboost, proposed by Freund et al. [10], is arguably the most popular boosting algorithm in practice. It aims to minimize the exponential loss, and has many variants which use some other surrogate loss. The main reason of using a surrogate loss is ease of optimization; while 0-1 loss is not even continuous, most surrogate losses are convex. We adopt the use of a surrogate loss for the same reason, and throughout this section will discuss our choice of surrogate loss for the adaptive algorithm.

Exponential loss is a very strong candidate in that it provides a closed form for computing potential functions, which are used to design cost matrices (cf. Mukherjee and Schapire [4, Theorem 13]). One property of online setting, however, makes it unfavorable. Like OnlineMBBM, each data point will have a different weight depending on weak learners' performance, and if the algorithm uses exponential loss, this weight will be an exponential function of difference in weighted cumulative votes. With this exponentially varying weights among samples, the algorithm might end up depending on very small portion of observed samples. This is undesirable because it is easier for the adversary to manipulate the sample sequence to perturb the learner.

To overcome exponentially varying weights, Beygelzimer et al. [7] use logistic loss in their adaptive algorithm. Logistic loss is more desirable in that its derivative is bounded and thus weights will be relatively smooth. For this reason, we will also use multiclass version of logistic loss:

$$L^r(\mathbf{s}) =: \sum_{l \neq r} \log(1 + \exp(\mathbf{s}[r] - \mathbf{s}[r])). \tag{8}$$

We still need to compute potential functions from logistic loss in order to calculate cost matrices. Unfortunately, Mukherjee and Schapire [4] use a unique property of exponential loss to get a closed form for potential functions, which cannot be adopted to logistic loss. However, the optimal cost matrix induced from exponential loss has a very close connection with the gradient of the loss (cf. Mukherjee and Schapire [4, Lemma 22]). From this, we will design our cost matrices as following:

$$\mathbf{C}_t^i[r, l] := \begin{cases} \frac{1}{1 + \exp(\mathbf{s}_t^{i-1}[r] - \mathbf{s}_t^{i-1}[l])} & \text{, if } l \neq r \\ -\sum_{j \neq r} \frac{1}{1 + \exp(\mathbf{s}_t^{i-1}[r] - \mathbf{s}_t^{i-1}[j])} & \text{, if } l = r. \end{cases} \tag{9}$$

Readers should note that the row vector $\mathbf{C}_t^i[r]$ is simply the gradient of $L^r(\mathbf{s}_t^{i-1})$. Also note that this matrix does not belong to $\mathcal{C}_1^{eor}$, but it does guarantee that the correct prediction gets the minimal cost.

The choice of logistic loss over exponential loss is somewhat subjective. The undesirable property of exponential loss does not necessarily mean that we cannot build an adaptive algorithm using this loss. In fact, we can slightly modify Algorithm 2 to develop algorithms using different surrogates (exponential loss and square hinge loss). However, their theoretical bounds are inferior to the one with logistic loss. Interested readers can refer Appendix D, but it assumes understanding of Algorithm 2.

## 4.2 Adaboost.OLM

Our work is a generalization of Adaboost.OL by Beygelzimer et al. [7], from which the name Adaboost.OLM comes with M standing for multiclass. We introduce a new concept of an *expert*. From $N$ weak learners, we can produce $N$ experts where expert $i$ makes its prediction by weighted majority votes among the first $i$ learners. Unlike OnlineMBBM, we allow varying weights $\alpha_t^i$ over the learners. As we are working with logistic loss, we want to minimize $\sum_t L^{y_t}(\mathbf{s}_t^i)$ for each $i$, where the loss is given in (8). We want to alert the readers to note that even though the algorithm tries to minimize the cumulative surrogate loss, its performance is still evaluated by 0-1 loss. The surrogate loss only plays a role of a bridge that makes the algorithm adaptive.

We do not impose the online weak learning condition on weak learners, but instead just measure the performance of $WL^i$ by $\gamma_i := \frac{\sum_t \mathbf{C}_t^i[y_t, l_t^i]}{\sum_t \mathbf{C}_t^i[y_t, y_t]}$. This *empirical edge* will be used to bound the number of mistakes made by Adaboost.OLM. By definition of cost matrix, we can check

$$\mathbf{C}_t^i[y_t, y_t] \leq \mathbf{C}_t^i[y_t, l] \leq -\mathbf{C}_t^i[y_t, y_t], \ \forall l \in [k],$$

from which we can prove $-1 \leq \gamma_i \leq 1, \ \forall i$. If the online weak learning condition is met with edge $\gamma$, then one can show that $\gamma_i \geq \gamma$ with high probability when the sample size is sufficiently large.

---

**Algorithm 2** Adaboost.OLM

---

1: **Initialize:** $\forall i, v_1^i = 1, \alpha_1^i = 0$
2: **for** $t = 1, \cdots, T$ **do**
3:     Receive example $\mathbf{x}_t$
4:     Set $\mathbf{s}_t^0 = \mathbf{0} \in \mathbb{R}^k$
5:     **for** $i = 1, \cdots, N$ **do**
6:         Compute $\mathbf{C}_t^i$ according to (9) and pass it to $WL^i$
7:         Set $l_t^i = WL^i(\mathbf{x}_t)$ and $\mathbf{s}_t^i = \mathbf{s}_t^{i-1} + \alpha_t^i \mathbf{e}_{l_t^i}$
8:         Set $\hat{y}_t^i = \operatorname{argmax}_l \mathbf{s}_t^i[l]$, the prediction of expert $i$
9:     **end for**
10:     Randomly draw $i_t$ with $\mathbb{P}(i_t = i) \propto v_t^i$
11:     Predict $\hat{y}_t = \hat{y}_t^{i_t}$ and receive the true label $y_t$
12:     **for** $i = 1, \cdots, N$ **do**
13:         Set $\alpha_{t+1}^i = \Pi(\alpha_t^i - \eta_t f_t^{i\,\prime}(\alpha_t^i))$ using (10) and $\eta_t = \frac{2\sqrt{2}}{(k-1)\sqrt{t}}$
14:         Set $\mathbf{w}^i[t] = -\frac{\mathbf{C}_t^i[y_t, y_t]}{k-1}$ and pass $(\mathbf{x}_t, y_t, \mathbf{w}^i[t])$ to $WL^i$
15:         Set $v_{t+1}^i = v_t^i \cdot \exp(-\mathbb{1}(y_t \neq \hat{y}_t^i))$
16:     **end for**
17: **end for**

---

Unlike the optimal algorithm, we cannot show the last expert that utilizes all the learners has the best accuracy. However, we can show at least one expert has a good predicting power. Therefore we will use classical *Hedge algorithm* (Littlestone and Warmuth [11] and Freund and Schapire [12]) to randomly choose an expert at each iteration with adaptive probability weight depending on each expert's prediction history.

Finally we need to address how to set the weight $\alpha_t^i$ for each weak learner. As our algorithm tries to minimize the cumulative logistic loss, we want to set $\alpha_t^i$ to minimize $\sum_t L^{y_t}(\mathbf{s}_t^{i-1} + \alpha_t^i \mathbf{e}_{l_t^i})$. This is again a classical topic in online learning, and we will use *online gradient descent*, proposed by Zinkevich [13]. By letting, $f_t^i(\alpha) := L^{y_t}(\mathbf{s}_t^{i-1} + \alpha \mathbf{e}_{l_t^i})$, we need an online algorithm ensuring $\sum_t f_t^i(\alpha_t^i) \leq \min_{\alpha \in F} \sum_t f_t^i(\alpha) + R^i(T)$ where $F$ is a feasible set to be specified later, and $R^i(T)$ is a regret that is sublinear in $T$. To apply Zinkevich [13, Theorem 1], we need $f_t^i$ to be convex and $F$ to be compact. The first assumption is met by our choice of logistic loss, and for the second assumption, we will set $F = [-2, 2]$. There is no harm to restrict the choice of $\alpha_t^i$ by $F$ because we can always scale the weights without affecting the result of weighted majority votes.

By taking derivatives, we get

$$f_t^{i\,\prime}(\alpha) = \begin{cases} \frac{1}{1+\exp(\mathbf{s}_t^{i-1}[y_t] - \mathbf{s}_t^{i-1}[l_t^i] - \alpha)} & \text{, if } l_t^i \neq y_t \\ -\sum_{j \neq y_t} \frac{1}{1+\exp(\mathbf{s}_t^{i-1}[j] + \alpha - \mathbf{s}_t^{i-1}[y_t])} & \text{, if } l_t^i = y_t. \end{cases} \tag{10}$$

This provides $|f_t^{i\,\prime}(\alpha)| \leq k - 1$. Now let $\Pi(\cdot)$ represent a projection onto $F$: $\Pi(\cdot) := \max\{-2, \min\{2, \cdot\}\}$. By setting $\alpha_{t+1}^i = \Pi(\alpha_t^i - \eta_t f_t^{i\,\prime}(\alpha_t^i))$ where $\eta_t = \frac{2\sqrt{2}}{(k-1)\sqrt{t}}$, we get $R^i(T) \leq 4\sqrt{2}(k-1)\sqrt{T}$. Readers should note that any learning rate of the form $\eta_t = \frac{c}{\sqrt{t}}$ would work, but our choice is optimized to ensure the minimal regret.

The pseudocode for Adaboost.OLM is presented in Algorithm 2. In fact, if we put $k = 2$, Adaboost.OLM has the same structure with Adaboost.OL. As in OnlineMBBM, the booster also needs to pass the weight along with labeled instance. According to (9), it can be inferred that the weight is proportional to $-\mathbf{C}_t^i[y_t, y_t]$.

### 4.3 Mistake bound and comparison to the optimal algorithm

Now we present our second main result that provides a mistake bound of Adaboost.OLM. The main structure of the proof is adopted from Beygelzimer et al. [7, Theorem 4] but in a generalized cost matrix framework. The proof appears in Appendix C.

**Theorem 5. (Mistake bound of Adaboost.OLM)** *For any $T$ and $N$, with probability $1 - \delta$, the number of mistakes made by Adaboost.OLM satisfies the following inequality:*

$$\sum_{t=1}^{T} \mathbb{1}(y_t \neq \hat{y}_t) \leq \frac{8(k-1)}{\sum_{i=1}^{N} \gamma_i^2} T + \tilde{O}(\frac{kN^2}{\sum_{i=1}^{N} \gamma_i^2}),$$

*where $\tilde{O}$ notation suppresses dependence on $\log \frac{1}{\delta}$.*

**Remark.** *Note that this theorem naturally implies Beygelzimer et al. [7, Theorem 4]. The difference in coefficients is due to different scaling of $\gamma_i$. In fact, their $\gamma_i$ ranges from $[-\frac{1}{2}, \frac{1}{2}]$.*

Now that we have established a mistake bound, it is worthwhile to compare the bound with the optimal boosting algorithm. Suppose the weak learners satisfy the weak learning condition (1) with edge $\gamma$. For simplicity, we will ignore the excess loss $S$. As we have $\gamma_i = \frac{\sum_t \mathbf{C}_t^i[y_t, l_t^i]}{\sum_t \mathbf{C}_t^i[y_t, y_t]} \geq \gamma$ with high probability, the mistake bound becomes $\frac{8(k-1)}{\gamma^2 N} T + \tilde{O}(\frac{kN}{\gamma^2})$. In order to achieve error rate $\epsilon$, Adaboost.OLM requires $N \geq \frac{8(k-1)}{\epsilon \gamma^2}$ learners and $T = \tilde{\Omega}(\frac{k^2}{\epsilon^2 \gamma^4})$ sample size. Note that OnlineMBBM requires $N = \Omega(\frac{1}{\gamma^2} \ln \frac{k}{\epsilon})$ and $T = \min\{\tilde{\Omega}(\frac{k^{5/2}}{\epsilon \gamma}), \tilde{\Omega}(\frac{k}{\epsilon \gamma^2})\}$. Adaboost.OLM is obviously suboptimal, but due to its adaptive feature, its performance on real data is quite comparable to that by OnlineMBBM.

## 5  Experiments

We compare the new algorithms to existing ones for online boosting on several UCI data sets, each with $k$ classes[1]. Table 1 contains some highlights, with additional results and experimental details in the Appendix E. Here we show both the average accuracy on the final 20% of each data set, as well as the average run time for each algorithm. Best decision tree gives the performance of the best of 100 online decision trees fit using the VFDT algorithm in Domingos and Hulten [14], which were used as the weak learners in all other algorithms, and Online Boosting is an algorithm taken from Oza [5]. Both provide a baseline for comparison with the new Adaboost.OLM and OnlineMBBM algorithms. Best MBBM takes the best result from running the OnlineMBBM with five different values of the edge parameter $\gamma$.

Despite being theoretically weaker, Adaboost.OLM often demonstrates similar accuracy and sometimes outperforms Best MBBM, which exemplifies the power of adaptivity in practice. This power comes from the ability to use diverse learners efficiently, instead of being limited by the strength of the weakest learner. OnlineMBBM suffers from high computational cost, as well as the difficulty of choosing the correct value of $\gamma$, which in general is unknown, but when the correct value of $\gamma$ is used it peforms very well. Finally in all cases Adaboost.OLM and OnlineMBBM algorithms outperform both the best tree and the preexisting Online Boosting algorithm, while also enjoying theoretical accuracy bounds.

Table 1: Comparison of algorithm accuracy on final 20% of data set and run time in seconds. Best accuracy on a data set reported in **bold**.

| Data sets | $k$ | Best decision tree | | Online Boosting | | Adaboost.OLM | | Best MBBM | |
|---|---|---|---|---|---|---|---|---|---|
| Balance | 3 | 0.768 | 8 | 0.772 | 19 | 0.754 | 20 | **0.821** | 42 |
| Mice | 8 | 0.608 | 105 | 0.399 | 263 | 0.561 | 416 | **0.695** | 2173 |
| Cars | 4 | 0.924 | 39 | 0.914 | 27 | **0.930** | 59 | 0.914 | 56 |
| Mushroom | 2 | 0.999 | 241 | **1.000** | 169 | **1.000** | 355 | **1.000** | 325 |
| Nursery | 4 | 0.953 | 526 | 0.941 | 302 | 0.966 | 735 | **0.969** | 1510 |
| ISOLET | 26 | 0.515 | 470 | 0.149 | 1497 | 0.521 | 2422 | **0.635** | 64707 |
| Movement | 5 | 0.915 | 1960 | 0.870 | 3437 | 0.962 | 5072 | **0.988** | 18676 |

**Acknowledgments**

We acknowledge the support of NSF under grants CAREER IIS-1452099 and CIF-1422157.

## Footnotes

[1]Codes are available at `https://github.com/yhjung88/OnlineBoostingWithVFDT`

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
