[Supplementary Material · supp.pdf]

## Appendix A    Link between batch and online weak learning conditions

Let us begin the section by introducing the weak learning condition in the batch setting. Mukherjee and Schapire [4] have identified necessary and sufficient condition for boostability. We will focus on a sufficient condition due to reasons of computational tractability. In the batch setting, the entire training set is revealed. Let $D := \{(\mathbf{x}_t, y_t) \mid t = 1, \cdots, T\}$ be the training set and define a family of cost matrices:

$$\mathcal{C}^{eor} := \{\mathbf{C} \in \mathbb{R}^{T \times k} \mid \forall t, \ \mathbf{C}[t, y_t] = \min_{l \in [k]} \mathbf{C}[t, l]\}.$$

The superscript "eor" stands for "edge-over-random." We warn the readers not to confuse $\mathcal{C}^{eor}$ with $\mathcal{C}_1^{eor}$. They both impose similar row constraints, but the matrices in these sets have different dimensions: $T \times k$ and $k \times k$ respectively. $\mathcal{C}_1^{eor}$ also has additional an normalization constraint. Note that $\mathcal{C}^{eor}$ provides one cost vector for an instance whereas $\mathcal{C}_1^{eor}$ provides a matrix. This is necessary because if an adversary passes only a vector to an online learner, then the learner can simply make the prediction which minimizes the cost. Furthermore, in the online boosting setting, the booster does not know the true label when it computes a cost matrix.

The authors prove that if a weak learning space $\mathcal{H}$ satisfies the condition described in Definition 6, then it is boostable, which means there exists a convex linear combination of hypotheses in $\mathcal{H}$ that perfectly classifies $D$.

**Definition 6.  (Batch setting weak learning condition, Mukherjee and Schapire [4])** *Suppose $D$ is fixed and $\mathcal{C}^{eor}$ is defined as above. A weak learning space $\mathcal{H}$ is said to satisfy weak learning condition $(\mathcal{C}^{eor}, \boldsymbol{U}_\gamma)$ if $\forall \boldsymbol{C} \in \mathcal{C}^{eor}$, one can find a weak hypothesis $h \in \mathcal{H}$ such that*

$$\sum_{t=1}^{T} \boldsymbol{C}[t, h(\boldsymbol{x}_t)] \leq \boldsymbol{C} \bullet \boldsymbol{U}_\gamma'. \tag{11}$$

Now we present how our online weak learning condition (Definition 1) is naturally derived from the batch setting counterpart (Definition 6). We extend the arguments of Beygelzimer et al. [7]. The batch setting condition (11) can be interpreted as making the following two implicit assumptions:

1. (Richness condition) For any $\mathbf{C} \in \mathcal{C}^{eor}$, there is some hypothesis $h \in \mathcal{H}$ such that

$$\sum_{t=1}^{T} \mathbf{C}[t, h(\mathbf{x}_t)] \leq \mathbf{C} \bullet \mathbf{U}_\gamma'.$$

2. (Agnostic learnability) For any $\mathbf{C} \in \mathcal{C}^{eor}$ and $\epsilon \in (0, 1)$, there is an algorithm which can compute a nearly optimal hypothesis $h \in \mathcal{H}$, i.e.

$$\sum_{t=1}^{T} \mathbf{C}[t, h(\mathbf{x}_t)] \leq \inf_{h' \in \mathcal{H}} \sum_{t=1}^{T} \mathbf{C}[t, h'(\mathbf{x}_t)] + \epsilon T.$$

For the online setting, we will keep the richness assumption with $\mathbf{C}$ being the matrix consisting of rows of $w_t \mathbf{C}_t[y_t]$, and the data being drawn by a fixed adversary. That is to say, it is the online richness condition that imposes a restriction on adversary because the condition cannot be met by any $\mathcal{H}$ with fully adaptive adversary. For example, suppose an adversary draws samples uniformly at random from the set $\{(\mathbf{x}, 1), \cdots, (\mathbf{x}, k)\}$ for some fixed $\mathbf{x} \in \mathcal{X}$. There does not exist weak learning space $\mathcal{H}$ that satisfies the online richness condition with this adversary. The agnostic learnability assumption is also replaced by online agnostic learnability assumption. We present online versions of the above two assumptions:

1'. (Online richness condition) For any sample length $T$, any sequence of labeled examples $\{(\mathbf{x}_t, y_t) \mid t = 1, \cdots, T\}$ generated by a fixed adversary, and any series of pairs of weight and cost matrix $\{(w_t, \mathbf{C}_t) \in [0, 1] \times \mathcal{C}_1^{eor} \mid t = 1, \cdots, T\}$, there is some hypothesis $h \in \mathcal{H}$ such that

$$\sum_{t=1}^{T} w_t \mathbf{C}_t[y_t, h(\mathbf{x}_t)] \leq \mathbf{C} \bullet \mathbf{U}_\gamma', \tag{12}$$

where $\mathbf{C} \in \mathbb{R}^{T \times k}$ consists of rows of $w_t \mathbf{C}_t[y_t]$.

2'. (Online agnostic learnability) For any sample length $T$, $\delta \in (0, 1)$, and for any adaptively chosen series of pairs of weight and cost matrix $\{(w_t, \mathbf{C}_t) \in [0, 1] \times \mathcal{C}_1^{eor} \mid t = 1, \cdots, T\}$, there is an online algorithm which can generate predictions $\hat{y}_t$ such that with probability $1 - \delta$,

$$\sum_{t=1}^{T} w_t \mathbf{C}_t[y_t, \hat{y}_t] \leq \inf_{h \in \mathcal{H}} \sum_{t=1}^{T} w_t \mathbf{C}_t[y_t, h(\mathbf{x}_t)] + R_\delta(T), \qquad (13)$$

where $R_\delta : \mathbb{N} \to \mathbb{R}$ is a sublinear regret.

Daniely et al. [15] extensively investigates agnostic learnability in online multiclass problems by introducing the following generalized Littlestone dimension (Littlestone [16]) of a hypothesis family $\mathcal{H}$. Consider a binary rooted tree $RT$ whose internal nodes are labeled by elements from $\mathcal{X}$ and whose edges are labeled by elements from $[k]$ such that two edges from a same parent have different labels. The tree $RT$ is *shattered* by $\mathcal{H}$ if, for every path from root to leaf which traverses the nodes $\mathbf{x}_1, \cdots, \mathbf{x}_k$, there is a hypothesis $h \in \mathcal{H}$ such that $h(\mathbf{x}_i)$ corresponds to the label of the edge from $\mathbf{x}_i$ to $\mathbf{x}_{i+1}$. The *Littlestone dimension* of $\mathcal{H}$ is the maximal depth of complete binary tree that is shattered by $\mathcal{H}$ (or $\infty$ if one can build a arbitrarily deep shattered tree). The authors prove that an optimal online algorithm has a sublinear regret under the expected (w.r.t. the randomness of the algorithm) 0-1 loss if Littlestone dimension of $\mathcal{H}$ is finite.

Similarly we prove in Lemma 7 that the condition (13) is satisfied if $\mathcal{H}$ has a finite Littlestone dimension. We need to slightly modify their result in two ways. One is to replace expectation by probabilistic argument, and the other is to replace 0-1 loss by our cost matrix framework. Both questions can be resolved by replacing an auxiliary lemma used by Daniely et al. [15] without changing the main structure.

**Lemma 7.** *Suppose a weak learning space $\mathcal{H}$ has a finite Littlestone dimension $d$ and an adversary chooses examples in fully adaptive manner. For any sample length $T$ and for any adaptively chosen series of pairs of weight and cost matrix $\{(w_t, \boldsymbol{C}_t) \in [0, 1] \times \mathcal{C}_1^{eor} \mid t = 1, \cdots, T\}$, with probability $1 - \delta$, the online agnostic learnability condition (13) is satisfied with following sublinear regret*

$$R_\delta(T) = \sqrt{(T d \ln T k)/2} + \sqrt{(T \ln 1/\delta)/2}.$$

*Proof.* We first introduce an online algorithm with experts. Suppose we have a fixed pool of experts of size $N$. We keep our cost matrix framework. Each expert $f^i$ would suffer cumulative cost $C_T^i := \sum_{t=1}^{T} w_t \mathbf{C}_t[y_t, f^i(\mathbf{x}_t)]$. At each iteration, an online algorithm chooses to follow one expert and incurs a cost $w_t \mathbf{C}_t[y_t, \hat{y}_t]$, and its goal is to perform as well as the best expert. That is to say, the algorithm wants to keep its cumulative cost $\sum_{t=1}^{T} w_t \mathbf{C}_t[y_t, \hat{y}_t]$ not too much larger than $\min_{i \in [N]} C_T^i$. This learning framework is called *weighted majority algorithm* and is thoroughly investigated by several researchers (e.g., Littlestone and Warmuth [11] and Vovk [17]). We will specifically use Algorithm 3 (LEA), which is shown to achieve a sublinear regret $\sqrt{(T \ln N)/2} + \sqrt{(T \ln 1/\delta)/2}$ with probability $1 - \delta$ (cf. Cesa-Bianchi and Lugosi [18, Corollary 4.2]). The authors require the loss to be bounded, which is also satisfied in our cost matrix framework. Readers might raise a question that our loss function changes for each iteration, but the proof still works as long as it is bounded. Interested readers might refer Hazan et al. [19, Section 1.3.3].

To apply this result in our case, we need to construct a finite set of experts whose best performance is as good as that of hypotheses in $\mathcal{H}$. In fact, in the proof of Daniely et al. [15, Theorem 25], the authors construct a set $E$ of size $N \leq (Tk)^d$ such that for every hypothesis $h \in \mathcal{H}$, there is an expert $f \in E$ which coincides with $h$ subject to the given examples $\mathbf{x}_1, \cdots, \mathbf{x}_T$.

Applying the LEA result on $E$ shows that with probability $1 - \delta$, the regret is bounded above by $\sqrt{(T d \ln T k)/2} + \sqrt{(T \ln 1/\delta)/2}$, which concludes the proof. $\qquad \square$

One remark is that the proof of Lemma 7 only uses the boundedness condition of $\mathcal{C}_1^{eor}$.

Now we are ready to demonstrate that our online weak learning condition is indeed naturally derived from the batch setting counterpart. The following Theorem shows that two conditions (12) and (13) directly imply the online weak learning condition (1). In other words, if the weak learning space $\mathcal{H}$ accompanied by an adversary is rich enough to contain a hypothesis that slightly outperforms a random guess and has a reasonably small dimension, then we can find an excess loss $S$ that satisfies

---

**Algorithm 3** Learning with Expert Advice (LEA)

---

1: **Input** T: time horizon, N: number of experts
2: Set $\eta = \sqrt{(8 \ln N)/T}$
3: Set $C_0^i = 0$ for all $i$
4: **for** $t = 1, \cdots, T$ **do**
5:     Receive example $\mathbf{x}_t$
6:     Receive expert advices $(f_t^1, \cdots, f_t^N) \in [k]^N$
7:     Predict $\hat{y}_t = f_t^i$ with probability proportional to $\exp(-\eta C_{t-1}^i)$
8:     Receive true label $y_t$
9:     Update $C_t^i = C_{t-1}^i + w_t \mathbf{C}_t[y_t, f_t^i]$ for all $i$
10: **end for**

---

(1). This is a generalization of Beygelzimer et al. [7, Lemma 2]. Note that we impose an additional assumption that $w_t \geq m > 0$ , $\forall t$. In case the learner encounters zero weight, it can simply ignore the instance, and the above assumption is not too artificial.

**Theorem 8. (Link between batch and online weak learning conditions)** *Suppose a pair of weak learning space $\mathcal{H}$ and an adversary satisfies online richness assumption (12) with edge $2\gamma$ and online agnostic learnability assumption (13) with mistake probability $\delta$ and sublinear regret $R_\delta(\cdot)$. Additionally we assume there exists a positive constant $m$ that satisfies $w_t \geq m$ , $\forall t$. Then the online learning algorithm satisfies the online weak learning condition (1), with mistake probability $\delta$, edge $\gamma$, and excess loss $S = \max_T(R_\delta(T) - \frac{\gamma m T}{k})$.*

*Proof.* Fix $\delta \in (0, 1)$ and a series of pairs of weight and cost matrix $\{(w_t, \mathbf{C}_t) \in [0, 1] \times \mathcal{C}_1^{eor} \mid t = 1, \cdots, T\}$, and let $\mathbf{C} \in \mathbb{R}^{T \times k}$ consist of rows of $w_t \mathbf{C}_t[y_t]$. First note that by sublinearity of $R_\delta(\cdot)$, $S$ is finite. According to (13), the online learning algorithm can generate predictions $\hat{y}_t$ such that, with probability $1 - \delta$,

$$\sum_{t=1}^{T} w_t \mathbf{C}_t[y_t, \hat{y}_t] \leq \mathbf{C} \bullet \mathbf{U}_{2\gamma}' + R_\delta(T).$$

Thus it suffices to show that

$$\mathbf{C} \bullet \mathbf{U}_{2\gamma}' + R_\delta(T) \leq \mathbf{C} \bullet \mathbf{U}_\gamma' + S. \tag{14}$$

Since the correct label gets zero cost and the row $\mathbf{C}[r]$ has $\ell_1$ norm $w_t$, we have

$$\mathbf{C} \bullet \mathbf{U}_\gamma' = \frac{1 - \gamma}{k} \|\mathbf{C}\|_1 = \frac{1 - \gamma}{k} \sum_{t=1}^{T} w_t.$$

By plugging this in (14), we get

$$\mathbf{C} \bullet \mathbf{U}_{2\gamma}' - \mathbf{C} \bullet \mathbf{U}_\gamma' + R_\delta(T) = -\frac{\gamma}{k} \sum_{t=1}^{T} w_t + R_\delta(T) \leq -\frac{\gamma}{k} m T + R_\delta(T) \leq S.$$

The first inequality holds because $w_t \geq m$, and the second inequality holds by definition of $S$, which completes the proof. $\square$

Lemma 7 and Theorem 8 suggest an implicit relation between $\delta$ and $S$ in (1). If we want probabilistically stronger weak learning condition, $R_\delta(T)$ in Lemma 7 gets bigger, which results in larger $S = \max_T(R_\delta(T) - \frac{\gamma T}{k})$.

# Appendix B   Detailed discussion of OnlineMBBM

## B.1   Proof of Theorem 2

*Proof.* For ease of notation, we will assume the edge is equal to $\gamma$ and the true label is $r$ unless otherwise specified. That is to say, $\mathbf{u}$ stands for $\mathbf{u}_\gamma^r$ and $\phi_i$ for $\phi_i^r$. By rewriting (3),

$$\begin{aligned}
\phi_{N-i+1}(\mathbf{s}_t^{i-1}) &= \mathbb{E}_{l\sim\mathbf{u}}\phi_{N-i}(\mathbf{s}_t^{i-1}+\mathbf{e}_l) \\
&= \mathbf{C}_t^i[r] \bullet \mathbf{u} \\
&= \mathbf{C}_t^i[r] \bullet (\mathbf{u}-\mathbf{e}_{l_t^i}) + \phi_{N-i}(\mathbf{s}_t^i),
\end{aligned}$$

where $\mathbf{C}_t^i$ is defined in (4). The last equation holds due to the relation $\mathbf{s}_t^i = \mathbf{s}_t^{i-1} + \mathbf{e}_{l_t^i}$. Also note that $||\mathbf{u}||_1 = ||\mathbf{e}_r||_1 = 1$, and thus subtracting common numbers from each component of $\mathbf{C}_t^i[r]$ does not affect the dot product term. Therefore, by introducing normalized cost matrix $\mathbf{D}_t^i$ as in (5) and $\mathbf{w}^i[t]$ as in Algorithm 1, we may write

$$\begin{aligned}
\phi_{N-i+1}^{y_t}(\mathbf{s}_t^{i-1}) &= \mathbf{w}^i[t]\mathbf{D}_t^i[y_t] \bullet (\mathbf{u}_\gamma^{y_t}-\mathbf{e}_{l_t^i}) + \phi_{N-i}^{y_t}(\mathbf{s}_t^i) \\
&= \mathbf{w}^i[t]\mathbf{D}_t^i[y_t] \bullet \mathbf{u}_\gamma^{y_t} - \mathbf{w}^i[t]\mathbf{D}_t^i[y_t,l_t^i] + \phi_{N-i}^{y_t}(\mathbf{s}_t^i) \qquad (15)\\
&= \mathbf{w}^i[t]\frac{1-\gamma}{k} - \mathbf{w}^i[t]\mathbf{D}_t^i[y_t,l_t^i] + \phi_{N-i}^{y_t}(\mathbf{s}_t^i).
\end{aligned}$$

The last equality holds because $\mathbf{D}_t^i$ is normalized and $\mathbf{D}_t^i[y_t,y_t]=0$. If $\mathbf{D}_t^i[y_t]$ is a zero vector, then by definition $\mathbf{w}^i[t]=0$, and the equality still holds. Then by summing (15) over $t$, we get

$$\sum_{t=1}^T \phi_{N-i+1}^{y_t}(\mathbf{s}_t^{i-1}) = \frac{1-\gamma}{k}||\mathbf{w}^i||_1 - \sum_{t=1}^T \mathbf{w}^i[t]\mathbf{D}_t^i[y_t,l_t^i] + \sum_{t=1}^T \phi_{N-i}^{y_t}(\mathbf{s}_t^i).$$

By online weak learning condition, we have with probability $1-\delta$, (recall that $w^{i*}$ estimates $||\mathbf{w}^i||_\infty$)

$$\sum_{t=1}^T \frac{\mathbf{w}^i[t]}{w^{i*}}\mathbf{D}_t^i[y_t,l_t^i] \le \frac{1-\gamma}{k}\frac{||\mathbf{w}^i||_1}{w^{i*}} + S.$$

From this, we can argue that

$$\sum_{t=1}^T \phi_{N-i+1}^{y_t}(\mathbf{s}_t^{i-1}) + Sw^{i*} \ge \sum_{t=1}^T \phi_{N-i}^{y_t}(\mathbf{s}_t^i).$$

Since the above inequality holds for any $i$, summing over $i$ gives

$$\sum_{t=1}^T \phi_N^{y_t}(\mathbf{0}) + S\sum_{i=1}^N w^{i*} \ge \sum_{t=1}^T \phi_0^{y_t}(\mathbf{s}_t^N),$$

which holds with probability $1-N\delta$ by union bound. By symmetry, $\phi_N^{y_t}(\mathbf{0}) = \phi_N^1(\mathbf{0})$ regardless of the true label $y_t$, and by definition of potential function (3), $\phi_0^{y_t}(\mathbf{s}_t^N) = L^{y_t}(\mathbf{s}_t^N)$, which completes the proof.

$\square$

## B.2   Bounding the terms in general bound under 0-1 loss

Even though OnlineMBBM has a promising theoretical justification, it would be infeasible if the computation of potential functions takes too long or if the behavior of asymptotic error rate $\phi_N^1(\mathbf{0})$ is too complicated to be approximated. Fortunately for the 0-1 loss, we can get a computationally tractable algorithm with vanishing error rate. The use of potential functions in binary boosting setup is thoroughly discussed by Schapire [20]. In binary setting under 0-1 loss, potential function has a closed form which dramatically reduces the computational complexity. Unfortunately, the multiclass

version does not have a closed form, but Mukherjee and Schapire [4] introduce a heuristic to compute it in reasonable time:

$$\phi_i^r(\mathbf{s}) = 1 - \sum_{(x_1, \cdots, x_k) \in A} \binom{i}{x_1, \cdots, x_k} \prod_{l=1}^{k} u_l^{x_l}, \tag{16}$$

where $A := \{(x_1, \cdots x_k) \in \mathbb{Z}^k \mid x_1 + \cdots x_k = i, \forall l : x_l \geq 0, x_l + \mathbf{s}[l] < x_r + \mathbf{s}[r]\}$, and $\mathbf{u}_\gamma^r = (u_1, \cdots, u_k)$. By using dynamic programming, the RHS of (16) can be computed in polynomial time in $i$, $k$, and $||\mathbf{s}||_1$. In our setting where the number of learners is fixed to be $N$, the computation can be done in polynomial time in $k$ and $N$ because $||\mathbf{s}||_1$ is bounded by $N$. To the best of our knowledge, there is no way to compute the potential function in polynomial time if we start from necessary and sufficient weak learning condition (the algorithm given by Mukherjee and Schapire [4] takes exponential time in the number of learners), and this is the main reason that we use the sufficient condition. Recall from (6) that $\phi_N^1(\mathbf{0})$ plays a role of asymptotic error rate and the second term determines the sample complexity. The following two lemmas provide bounds for both terms.

By applying the Hoeffding's inequality, we can prove in Lemma 9 that $\phi_N^1(\mathbf{0})$ vanishes exponentially fast as $N$ grows. That is to say, to get a satisfactory accuracy, we do not need too many learners. We also note that we can decide $N$ before the learning process begins, which is logically plausible.

**Lemma 9.** *Under the same setting as in Theorem 2 but with the particular choice of 0-1 loss, we may bound $\phi_N^1(\mathbf{0})$ as follows:*

$$\phi_N^1(\mathbf{0}) \leq (k-1)\exp(-\frac{\gamma^2 N}{2}). \tag{17}$$

*Proof.* We reinterpret $\phi_N^1(\mathbf{0})$ in (16). Imagine that we draw numbers $N$ times from $[k]$ where the probability that a number $i$ is drawn is $\mathbf{u}_\gamma^1[i]$. That is to say, 1 has highest probability of $\frac{1-\gamma}{k} + \gamma$, and other numbers have equal probability of $\frac{1-\gamma}{k}$. Then $\phi_N^1(\mathbf{0})$ can be interpreted as a probability that the number that is drawn for the most time out of $N$ draws is not 1. Let $A_i$ denote the event that the number $i$ gets more votes than the number 1. Then we have by union bound,

$$\phi_N^1(\mathbf{0}) = \mathbb{P}(A_2 \cup \cdots \cup A_k)$$
$$\leq \sum_{l=2}^{k} \mathbb{P}(A_i) \tag{18}$$
$$= (k-1)\mathbb{P}(A_2)$$

The last equality holds by symmetry. To compute $\mathbb{P}(A_2)$, imagine that we draw 1 with probability $\frac{1-\gamma}{k} + \gamma$, $-1$ with probability $\frac{1-\gamma}{k}$, and 0 otherwise. $\mathbb{P}(A_2)$ is equal to the probability that after independent $N$ draws, the summation of $N$ i.i.d. random numbers is non-positive. Thus by the Hoeffding's inequality, we get

$$\mathbb{P}(A_2) \leq \exp(-\frac{\gamma^2 N}{2}) \tag{19}$$

Combining (18) and (19) completes the proof. □

Now we have fixed $N$ based on the desired asymptotic accuracy. Since 0-1 loss is bounded in $[0, 1]$, so are potential functions. Then by definition of weights (cf. Algorithm 1), $||\mathbf{w}^i||_\infty$ is trivially bounded above by $k$, which means we can use $w^{i*} = k \ \forall i$. Thus the second term of (6) is bounded above by $kNS$, which is valid. However, Lemma 10 allows a tighter bound.

**Lemma 10.** *Under the same setting as in Theorem 2 but with the particular choice of 0-1 loss and an additional constraint of $\gamma < \frac{1}{2}$, we may bound $||\mathbf{w}^i||_\infty$ by*

$$||\mathbf{w}^i||_\infty \leq \frac{ck^{5/2}}{\sqrt{N-i}}, \tag{20}$$

*where $c$ is a universal constant that can be determined before the algorithm begins.*

*Proof.* We will start by providing a bound on $\phi_m^r(\mathbf{s} + \mathbf{e}_l) - \phi_m^r(\mathbf{s} + \mathbf{e}_r)$. First note that it is non-negative as potential functions are proper. Again by using random draw framework as in the proof of Lemma 9 (now $r$ has the largest probability to be drawn), this value corresponds to the probability that after $m$ draws, the number $r$ wins the majority votes if the count starts from $\mathbf{s} + \mathbf{e}_r$ but loses if the count starts from $\mathbf{s} + \mathbf{e}_l$. Let $X_1, \cdots, X_k$ denote the number of draws of each number out of $m$ draws and define the events $A_l := \{(X_r + \mathbf{s}[r]) - (X_l + \mathbf{s}[l]) \in \{0, 1\}\}$. Then it can be checked that

$$
\begin{aligned}
&\phi_m^r(\mathbf{s} + \mathbf{e}_l) - \phi_m^r(\mathbf{s} + \mathbf{e}_r) \\
&= \mathbb{P}(\exists l' \text{ s.t. } X_{l'} + \mathbf{s}[l'] + \mathbf{e}_l[l'] \geq X_r + \mathbf{s}[r]) - \mathbb{P}(\exists l' \text{ s.t. } X_{l'} + \mathbf{s}[l'] \geq X_r + \mathbf{s}[r] + 1) \\
&\leq \mathbb{P}(\exists l' \text{ s.t. } X_{l'} + \mathbf{s}[l'] + \mathbf{e}_l[l'] \geq X_r + \mathbf{s}[r] \text{ and } \forall l', X_r + \mathbf{s}[r] \geq X_{l'} + \mathbf{s}[l']) \\
&\leq \mathbb{P}(\exists l' \text{ s.t. } X_{l'} + \mathbf{s}[l'] + \mathbf{e}_l[l'] \geq X_r + \mathbf{s}[r] \geq X_{l'} + \mathbf{s}[l']) \\
&= \mathbb{P}(\bigcup_{l \neq r} A_l) \leq \sum_{l \neq r} \mathbb{P}(A_l).
\end{aligned}
\tag{21}
$$

The first inequality holds by $\mathbb{P}(A) - \mathbb{P}(B) \leq \mathbb{P}(A - B)$. Individual probabilities can be written as

$$
\begin{aligned}
\mathbb{P}(A_l) &= \mathbb{P}(X_r - X_l = \mathbf{s}[l] - \mathbf{s}[r]) + \mathbb{P}(X_r - X_l = \mathbf{s}[l] - \mathbf{s}[r] + 1) \\
&\leq 2 \max_n \mathbb{P}(X_r - X_l = n).
\end{aligned}
\tag{22}
$$

We can prove by applying the Berry-Esseen theorem that the last probability is $O(\frac{1}{\sqrt{m}})$. Let $Y_1, \cdots, Y_m$ be a sequence of i.i.d. random variables such that $Y_j \in \{-1, 0, 1\}$ and

$$
\mathbb{P}(Y_j = 1) = \frac{1 - \gamma}{k} + \gamma,
$$

$$
\mathbb{P}(Y_j = -1) = \frac{1 - \gamma}{k}.
$$

Note that $\mathbb{E}Y_j = \gamma$ and $Var(Y_j) = \frac{2(1-\gamma)}{k} + \gamma(1 - \gamma) =: \sigma^2$. It can be easily checked that $Y := \sum_{j=1}^m Y_j$ has same distribution with $X_r - X_l$. Now we approximate $Y$ by a Gaussian random variable $W \sim N(m\gamma, m\sigma^2)$. Let $F_W$ and $F_Y$ denote CDF of $W$ and $Y$, respectively, and let $f$ denote the density of $W$. First note that

$$
|\mathbb{P}(Y = n) - \int_{n-1}^n f(w)dw| = |(F_Y(n) - F_Y(n - 1)) - (F_W(n) - F_W(n - 1))|
$$

$$
\leq |F_Y(n) - F_W(n)| + |F_Y(n - 1) - F_W(n - 1)|.
$$

We can apply the Berry-Esseen theorem to the last CDF differences, which provides

$$
|\mathbb{P}(Y = n) - \int_{n-1}^n f(w)dw| \leq \frac{2C\rho}{\sigma^3\sqrt{m}},
\tag{23}
$$

where $C$ is the universal constant that appears in Berry-Esseen and $\rho := \mathbb{E}|Y_j - \gamma|^3$. As $Y_j$ is a bounded random variable, we have

$$
\rho = \mathbb{E}|Y_j - \gamma|^3 \leq (1 + \gamma)\mathbb{E}|Y_j - \gamma|^2 = (1 + \gamma)\sigma^2 \leq 2\sigma^2.
$$

Plugging this in (23) gives

$$
|\mathbb{P}(Y = n) - \int_{n-1}^n f(w)dw| \leq \frac{4C}{\sigma\sqrt{m}}
$$

By simple algebra, we can deduce

$$
\begin{aligned}
\mathbb{P}(Y = n) &\leq \int_{n-1}^n f(w)dw + \frac{4C}{\sigma\sqrt{m}} \\
&\leq \sup_{w \in \mathbb{R}} f(w) + \frac{4C}{\sigma\sqrt{m}} \\
&= \frac{1}{\sqrt{2\pi m}\sigma} + \frac{4C}{\sigma\sqrt{m}}.
\end{aligned}
\tag{24}
$$

Using the fact that $\gamma < \frac{1}{2}$, we can show

$$\sigma^2 = \frac{2(1-\gamma)}{k} + \gamma(1-\gamma) \geq \frac{1}{k}$$

Plugging this in (24) gives

$$\mathbb{P}(Y = n) \leq \frac{1}{\sigma\sqrt{m}}(\frac{1}{\sqrt{2\pi}} + 4C) \leq C'\sqrt{\frac{k}{m}}, \qquad (25)$$

where $C' = \frac{1}{\sqrt{2\pi}} + 4C$. By combining (21), (22), (25), and the fact that $Y$ and $X_r - X_l$ have same distribution, we prove

$$\phi_m^r(\mathbf{s} + \mathbf{e}_l) - \phi_m^r(\mathbf{s} + \mathbf{e}_r) \leq 2C'k\sqrt{\frac{k}{m}}. \qquad (26)$$

The proof is complete by observing that $\mathbf{w}^i[t] = \sum_{l=1}^k [\phi_{N-i}^{y_t}(\mathbf{s}_t^{i-1} + \mathbf{e}_l) - \phi_{N-i}^{y_t}(\mathbf{s}_t^{i-1} + \mathbf{e}_{y_t})]$. $\quad\square$

**Remark.** *By summing (20) over $i$, we can bound the second term of (6) by $O(k^{5/2}\sqrt{N})S$. Comparing this to the aforementioned bound $kNS$, Lemma 10 reduces the dependency on $N$, but as a tradeoff the dependency on $k$ is increased. The optimal bound for this term remains open, but in the case that the number of classes $k$ is fixed to be moderate, Lemma 10 provides a better bound.*

Corollary 3 is a simple consequence of plugging Lemma 9 and 10 to Theorem 2.

### B.3 Proof of lower bounds and discussion of gap

We begin by proving Theorem 4.

*Proof.* At time $t$, an adversary draws a label $y_t$ uniformly at random from $[k]$, and the weak learners independently make predictions with respect to the probability distribution $\mathbf{p}_t \in \Delta[k]$. This can be achieved if the adversary draws $\mathbf{x}_t \in \mathbb{R}^N$ where $\mathbf{x}_t[1], \cdots, \mathbf{x}_t[N]|y_t$'s are conditionally independent with conditional distribution of $\mathbf{p}_t$ and $WL^i$ predicts $\mathbf{x}_t[i]$. The booster can only make a final decision by weighted majority votes of $N$ weak learners. We will manipulate $\mathbf{p}_t$ in such a way that weak learners satisfy (1), but the booster's performance is close to that of Online MBBM.

First we note that since $\mathbf{C}_t[y_t, \hat{y}_t]$ used in (1) is bounded in $[0, 1]$, the Azuma-Hoeffding inequality implies that if a weak learner makes prediction $\hat{y}_t$ according to the probability distribution $\mathbf{p}_t$ at time $t$, then with probability $1 - \delta$, we have

$$\begin{aligned}
\sum_{t=1}^T w_t \mathbf{C}_t[y_t, \hat{y}_t] &\leq \sum_{t=1}^T w_t \mathbf{C}_t[y_t] \bullet \mathbf{p}_t + \sqrt{2\|\mathbf{w}\|_2^2 \ln(\frac{1}{\delta})} \\
&\leq \sum_{t=1}^T w_t \mathbf{C}_t[y_t] \bullet \mathbf{p}_t + \frac{\gamma\|\mathbf{w}\|_2^2}{k} + \frac{k\ln(\frac{1}{\delta})}{2\gamma} \\
&\leq \sum_{t=1}^T w_t \mathbf{C}_t[y_t] \bullet \mathbf{p}_t + \frac{\gamma\|\mathbf{w}\|_1}{k} + \frac{k\ln(\frac{1}{\delta})}{2\gamma},
\end{aligned} \qquad (27)$$

where the second inequality holds by arithmetic mean and geometric mean relation and the last inequality holds due to $w_t \in [0, 1]$.

We start from providing a lower bound on the number of weak learners. Let $\mathbf{p}_t = \mathbf{u}_{2\gamma}^{y_t}$ for all $t$. This can be done by the constraint $\gamma < \frac{1}{4}$. Then the last line of (27) becomes

$$\sum_{t=1}^T w_t \mathbf{C}_t[y_t] \bullet \mathbf{u}_{2\gamma}^{y_t} + \frac{\gamma\|\mathbf{w}\|_1}{k} + \frac{k\ln(\frac{1}{\delta})}{2\gamma} = \frac{1-2\gamma}{k}\|\mathbf{w}\|_1 + \frac{\gamma\|\mathbf{w}\|_1}{k} + \frac{k\ln(\frac{1}{\delta})}{2\gamma} \leq \frac{1-\gamma}{k}\|\mathbf{w}\|_1 + S,$$

where the first equality follows by the fact that $\mathbf{C}_t[y_t, y_t] = 0$ and $\|\mathbf{C}_t[y_t]\|_1 = 1$. Thus the weak learners indeed satisfy the online weak learning condition with edge $\gamma$ and excess loss $S$. Now

suppose a booster imposes weights on weak learners by $\alpha^i$. WLOG, we may assume the weights are normalized such that $\sum_{i=1}^{N} \alpha^i = 1$. Adopting the argument of Schapire and Freund [3, Section 13.2.6], we prove that the optimal choice of weights is $(\frac{1}{N}, \cdots, \frac{1}{N})$. Fix $t$, and let $l^i$ denote the prediction made by $WL^i$. By noting that $\mathbb{P}(y_t = y) = \frac{1}{k}$, which is constant, we can deduce

$$\mathbb{P}(y_t = y | l^1, \cdots, l^N) = \frac{\mathbb{P}(l^1, \cdots, l^N | y_t = y)\mathbb{P}(y_t = y)}{\mathbb{P}(l^1, \cdots, l^N)}$$
$$\propto \mathbb{P}(l^1, \cdots, l^N | y_t = y)$$
$$= \prod_{i=1}^{N} p^{\mathbb{1}(l^i = y)} q^{\mathbb{1}(l^i \neq y)},$$

where $f \propto g$ means $f(y)/g(y)$ does not depend on $y$, $p = \mathbf{u}_{2\gamma}^{y_t}[y_t] = \frac{1-2\gamma}{k} + 2\gamma$, and $q = \mathbf{u}_{2\gamma}^{y_t}[l] = \frac{1-2\gamma}{k}$. By taking log, we get

$$\log \mathbb{P}(y_t = y | l^1, \cdots, l^N) = C + \log p \sum_{i=1}^{N} \mathbb{1}(l^i = y) + \log q \sum_{i=1}^{N} \mathbb{1}(l^i \neq y)$$
$$= C + N \log q + \log \frac{p}{q} \sum_{i=1}^{N} \mathbb{1}(l^i = y).$$

Therefore, the optimal decision after observing $l^1, \cdots, l^N$ is to choose $y$ that maximizes $\sum_{i=1}^{N} \mathbb{1}(l^i = y)$, or equivalently, to take simple majority votes.

To compute a lower bound for the error rate, we again introduce random draw framework as in the proof of Lemma 9. WLOG, we may assume that the true label is 1. Let $A_i$ denote the event that the number $i$ beats 1 in the majority votes. Then we have

$$\mathbb{P}(\text{booster makes error}) \geq \mathbb{P}(A_2). \tag{28}$$

Now we need a lower bound for $\mathbb{P}(A_2)$. To do so, let $\{Y_i\}$ be the series of i.i.d. random variables such that $Y_i \in \{-1, 0, 1\}$ and

$$\mathbb{P}(Y_j = 1) = \frac{1 - 2\gamma}{k} + 2\gamma =: p_1,$$
$$\mathbb{P}(Y_j = -1) = \frac{1 - 2\gamma}{k} =: p_{-1}.$$

Then $\mathbb{P}(A_2) = \mathbb{P}(Y < 0)$ where $Y := \sum_{i=1}^{N} Y_i$.

Now let $M$ be the number of $j$ such that $Y_j \neq 0$. By conditioning on $M$, we can write

$$\mathbb{P}(Y < 0 | M = m) = \mathbb{P}(B \leq \frac{m}{2}),$$

where $B \sim binom(m, \frac{p_1}{p_1 + p_{-1}})$. By Slud's inequality [21, Theorem 2.1], we have

$$\mathbb{P}(B \leq \frac{m}{2}) \geq \mathbb{P}(Z \geq \sqrt{m} \frac{p - \frac{1}{2}}{\sqrt{p(1-p)}}),$$

where $Z$ follows a standard normal distribution and $p = \frac{p_1}{p_1 + p_{-1}}$. Now using tail bound on normal distribution, we get

$$\mathbb{P}(B \leq \frac{m}{2}) \geq \Omega(\exp(-\frac{m(p - 1/2)^2}{p(1-p)}))$$
$$= \Omega(\exp(-\frac{m(p_1 - p_{-1})^2}{4p_1 p_{-1}}))$$
$$= \Omega(\exp(-\frac{m\gamma^2}{p_1 p_{-1}})) \tag{29}$$
$$\geq \Omega(\exp(-4mk^2\gamma^2))$$
$$\geq \Omega(\exp(-4Nk^2\gamma^2)).$$

Figure 1: Plot of $\phi_N^1(\mathbf{0})$ computed with distribution $\mathbf{u}_\gamma^1$ versus the number of labels $k$. $N$ is fixed to be 20, and the edge $\gamma$ is set to be 0.01 (left) and 0.1 (right). The graph is not monotonic for larger edge. This hinders the approximation of potential functions with respect to $k$.

We intentionally drop $\frac{1}{2}$ from the power, which makes the bound smaller. The second inequality holds because $p_1 p_{-1} \geq \frac{(1-2\gamma)^2}{k^2} \geq \frac{1}{4k^2}$. Integrating w.r.t. $m$ gives

$$\mathbb{P}(\text{booster makes error}) \geq \mathbb{P}(Y < 0) \geq \Omega(\exp(-4Nk^2\gamma^2)).$$

By setting this value equal to $\epsilon$, we have $N \geq \Omega(\frac{1}{k^2\gamma^2} \ln \frac{1}{\epsilon})$, which proves the first part of the theorem.

Now we turn our attention to the optimality of sample complexity. Let $T_0 := \frac{kS}{4\gamma}$ and define $\mathbf{p}_t = \mathbf{u}_0^{y_t}$ for $t \leq T_0$ and $\mathbf{p}_t = \mathbf{u}_{2\gamma}^{y_t}$ for $t > T_0$. Then for $T \leq T_0$, (27) implies

$$\sum_{t=1}^{T} w_t \mathbf{C}_t[y_t, \hat{y}_t] \leq \frac{1+\gamma}{k}||\mathbf{w}||_1 + \frac{k \ln(\frac{1}{\delta})}{2\gamma} \leq \frac{1-\gamma}{k}||\mathbf{w}||_1 + S, \tag{30}$$

where the last inequality holds because $||\mathbf{w}||_1 \leq T_0 = \frac{kS}{4\gamma}$. For $T > T_0$, again (27) implies

$$\begin{aligned}
\sum_{t=1}^{T} w_t \mathbf{C}_t[y_t, \hat{y}_t] &\leq \frac{1}{k}\sum_{t=1}^{T_0} w_t + \frac{1-2\gamma}{k}\sum_{t=T_0+1}^{T} w_t + \frac{\gamma||\mathbf{w}||_1}{k} + \frac{k \ln(\frac{1}{\delta})}{2\gamma} \\
&\leq \frac{2\gamma}{k}T_0 + \frac{1-\gamma}{k}||\mathbf{w}||_1 + \frac{k \ln(\frac{1}{\delta})}{2\gamma} \\
&\leq \frac{1-\gamma}{k}||\mathbf{w}||_1 + S.
\end{aligned} \tag{31}$$

(30) and (31) prove that the weak learners indeed satisfy (1). Now note that combining weak learners does not provide meaningful information for $t \leq T_0$, and thus any online boosting algorithm has errors at least $\Omega(T_0)$. Therefore to get the desired asymptotic error rate, the number of observations $T$ should be at least $\Omega(\frac{T_0}{\epsilon}) = \Omega(\frac{k}{\epsilon\gamma}S)$, which proves the second part of the theorem. $\qquad\square$

Even though the gap for the number of weak learners between Corollary 3 and Theorem 4 is merely polynomial in $k$, readers might think it is counter-intuitive that $N$ is increasing in $k$ in the upper bound while decreasing in the lower bound. This phenomenon occurs due to the difficulty in approximating potential functions. Recall that Lemma 9 and Theorem 4 utilize upper and lower bound of $\phi_N^1(\mathbf{0})$.

At first glance, considering that $\phi_N^1(\mathbf{0})$ implies the error rate of majority votes out of $N$ independent random draws with distribution $\mathbf{u}_\gamma^1$, the potential function seems to be increasing in $k$ as the task gets harder with bigger set of options. This is the case of left panel of Figure 1. However, as it is shown in the right panel, it can also start decreasing in $k$ when $\gamma$ is larger. This can happen because the probability that a wrong label is drawn vanishes as $k$ grows while the probability that the correct

label is drawn remains bigger than $\gamma$. In this regard, even though the number of wrong labels gets larger, the error rate actually decreases as $\mathbf{u}_\gamma^1[1]$ dominates other probabilities.

After acknowledging that $\phi_N^1(\mathbf{0})$ might not be a monotonic function of $k$, the linear upper bound (17) turns out to be quite naive, and this is the main reason for the conflicting dependence on $k$ in upper bound and lower bound for $N$. As the relation among $k$, $N$, and $\gamma$ in $\phi_N^1(\mathbf{0})$ is quite intricate, the issue of deriving better approximation of potential functions remains open.

## Appendix C   Proof of Theorem 5

We first introduce a lemma that will be used in the proof.

**Lemma 11.** *Suppose* $A, B \geq 0$, $B - A = \gamma \in [-1, 1]$, *and* $A + B \leq 1$. *Then we have*

$$\min_{\alpha \in [-2,2]} A(e^\alpha - 1) + B(e^{-\alpha} - 1) \leq -\frac{\gamma^2}{2}.$$

*Proof.* We divide into three cases with respect to the range of $\frac{B}{A}$.

First suppose $e^{-4} \leq \frac{B}{A} \leq e^4$. In this case, the minimum is attained at $\alpha = \frac{1}{2} \log \frac{B}{A}$, and the minimum becomes

$$
\begin{aligned}
-(A + B) + 2\sqrt{AB} &= -(\sqrt{A} - \sqrt{B})^2 \\
&= -(\frac{A - B}{\sqrt{A} + \sqrt{B}})^2 \\
&= -\frac{\gamma^2}{(\sqrt{A} + \sqrt{B})^2} \\
&\leq -\frac{\gamma^2}{2(A + B)} \leq -\frac{\gamma^2}{2}.
\end{aligned}
$$

Now suppose $\frac{B}{A} > e^4 > 51$. From $B - A = \gamma$, we have $\gamma > 50A \geq 0$. Choosing $\alpha = \log 6$, we get the minimum is bounded above by

$$
\begin{aligned}
5A - \frac{5}{6}B &= \frac{25}{6}A - \frac{5}{6}\gamma \\
&< \frac{25}{6}\frac{\gamma}{50} - \frac{5}{6}\gamma \\
&= -\frac{3}{4}\gamma < -\frac{\gamma^2}{2}.
\end{aligned}
$$

The last inequality hold due to $\gamma \leq 1$.

Finally suppose $\frac{A}{B} > e^4 > 51$. From $B - A = \gamma$, we have $-\gamma > 50B \geq 0$. Choosing $\alpha = -\log 6$, we get the minimum is bounded above by

$$
\begin{aligned}
-\frac{5}{6}A + 5B &= \frac{25}{6}B + \frac{5}{6}\gamma \\
&< -\frac{25}{6}\frac{\gamma}{50} + \frac{5}{6}\gamma \\
&= \frac{3}{4}\gamma < -\frac{\gamma^2}{2}.
\end{aligned}
$$

The last inequality hold due to $\gamma \geq -1$. This completes the proof. $\qquad\square$

Now we provide a proof of Theorem 5.

*Proof.* Let $M_i$ denote the number of mistakes made by expert $i$: $M_i = \sum_t \mathbb{1}(y_t \neq \hat{y}_t^i)$. We also let $M_0 = T$ for the ease of presentation. As Adaboost.OLM is using the Hedge algorithm among $N$

experts, the Azuma-Hoeffding inequality and a standard analysis (cf. Cesa-Bianchi and Lugosi [18, Corollary 2.3]) provide with probability $1 - \delta$,

$$\sum_t \mathbb{1}(y_t \neq \hat{y}_t) \leq 2 \min_i M_i + 2 \log N + \tilde{O}(\sqrt{T}), \tag{32}$$

where $\tilde{O}$ notation suppresses dependence on $\log \frac{1}{\delta}$.

Now suppose the expert $i - 1$ makes a mistake at iteration $t$. That is to say, in a conservative way, $\mathbf{s}_t^{i-1}[y_t] \leq \mathbf{s}_t^{i-1}[l]$ for some $l \neq y_t$. This implies that among $k - 1$ terms in the summation of $-\mathbf{C}_t^i[y_t, y_t]$ in (9), at least one term is not less than $\frac{1}{2}$. Thus we can say $-\mathbf{C}_t^i[y_t, y_t] \geq \frac{1}{2}$ if the expert $i - 1$ makes a mistake at $\mathbf{x}_t$. This leads to the inequality:

$$-\sum_t \mathbf{C}_t^i[y_t, y_t] \geq \frac{M_{i-1}}{2}. \tag{33}$$

Note that by definition of $M_0$ and $\mathbf{C}_t^1$, the above inequality holds for $i = 1$ as well. For ease of notation, let us write $w^i := -\sum_t \mathbf{C}_t^i[y_t, y_t]$.

Now let $\Delta_i$ denote the difference of the cumulative logistic loss between two consecutive experts:

$$\Delta_i = \sum_t L^{y_t}(\mathbf{s}_t^i) - L^{y_t}(\mathbf{s}_t^{i-1}) = \sum_t L^{y_t}(\mathbf{s}_t^{i-1} + \alpha_t^i \mathbf{e}_{l_t^i}) - L^{y_t}(\mathbf{s}_t^{i-1}).$$

Then Online Gradient Descent algorithm provides

$$\Delta_i \leq \min_{\alpha \in [-2,2]} \sum_t [L^{y_t}(\mathbf{s}_t^{i-1} + \alpha \mathbf{e}_{l_t^i}) - L^{y_t}(\mathbf{s}_t^{i-1})] + 4\sqrt{2}(k-1)\sqrt{T}. \tag{34}$$

By simple algebra, we can check

$$\log(1 + e^{s+\alpha}) - \log(1 + e^s) = \log(1 + \frac{e^\alpha - 1}{1 + e^{-s}}) \leq \frac{1}{1 + e^{-s}}(e^\alpha - 1).$$

From this, we can deduce that

$$L^{y_t}(\mathbf{s}_t^{i-1} + \alpha \mathbf{e}_{l_t^i}) - L^{y_t}(\mathbf{s}_t^{i-1}) \leq \begin{cases} \mathbf{C}_t^i[y_t, l_t^i](e^\alpha - 1) & , \text{if } l_t^i \neq y_t \\ \mathbf{C}_t^i[y_t, l_t^i](-e^{-\alpha} + 1) & , \text{if } l_t^i = y_t \end{cases}.$$

Summing over $t$, we have

$$\sum_t L^{y_t}(\mathbf{s}_t^{i-1} + \alpha \mathbf{e}_{l_t^i}) - L^{y_t}(\mathbf{s}_t^{i-1}) \leq w^i(A(e^\alpha - 1) + B(e^{-\alpha} - 1)),$$

where

$$A = \sum_{l_t \neq y_t} \mathbf{C}_t[y_t, l_t]/w^i, \ B = -\sum_{l_t = y_t} \mathbf{C}_t[y_t, l_t]/w^i.$$

Note that $A$ and $B$ are non-negative and $B - A = \gamma_i \in [-1, 1]$, $A + B \leq 1$. Lemma 11 provides

$$\min_{\alpha \in [-2,2]} \sum_t [L^{y_t}(\mathbf{s}_t^{i-1} + \alpha \mathbf{e}_{l_t^i}) - L^{y_t}(\mathbf{s}_t^{i-1})] \leq -\frac{\gamma_i^2}{2} w^i. \tag{35}$$

Combining (33), (34), and (35), we have

$$\Delta_i \leq -\frac{\gamma_i^2}{4} M_{i-1} + 4\sqrt{2}(k-1)\sqrt{T}.$$

Summing over $i$, we get by telescoping rule

$$\sum_t L^{y_t}(\mathbf{s}_t^N) - \sum_t L^{y_t}(\mathbf{0}) \leq -\frac{1}{4} \sum_i \gamma_i^2 M_{i-1} + 4\sqrt{2}(k-1)N\sqrt{T}$$

$$\leq -\frac{1}{4} \sum_i \gamma_i^2 \min_i M_i + 4\sqrt{2}(k-1)N\sqrt{T}.$$

Note that $L^{y_t}(\mathbf{0}) = (k-1)\log 2$ and $L^{y_t}(\mathbf{s}_t^N) \geq 0$. Therefore we have

$$\min_i M_i \leq \frac{4(k-1)\log 2}{\sum_i \gamma_i^2}T + \frac{16\sqrt{2}(k-1)N}{\sum_i \gamma_i^2}\sqrt{T}.$$

Plugging this in (32), we get with probability $1 - \delta$,

$$\sum_t \mathbb{1}(y_t \neq \hat{y}_t) \leq \frac{8(k-1)\log 2}{\sum_i \gamma_i^2}T + \tilde{O}\left(\frac{kN\sqrt{T}}{\sum_i \gamma_i^2} + \log N\right)$$

$$\leq \frac{8(k-1)}{\sum_i \gamma_i^2}T + \tilde{O}\left(\frac{kN^2}{\sum_i \gamma_i^2}\right),$$

where the last inequality holds from AM-GM inequality: $cN\sqrt{T} \leq \frac{c^2 N^2 + T}{2}$.

$\square$

## Appendix D  Adaptive algorithms with different surrogate losses

In this section, we present similar adaptive boosting algorithms with Adaboost.OLM but with two different surrogate losses: exponential loss and square hinge loss. We keep the main structure, but the unique properties of each loss result in little difference in details.

### D.1  Exponential loss

As discussed in Section 4.1, exponential loss is useful in batch setting because it provides a closed form for the potential function. We will use following multiclass version of exponential loss:

$$L^r(\mathbf{s}) := \sum_{l \neq r} \exp(\mathbf{s}[l] - \mathbf{s}[r]). \tag{36}$$

From this, we can compute the cost matrix and $f_t^{i'}$ for the online gradient descent as below:

$$\mathbf{C}_t^i[r, l] = \begin{cases} \exp(\mathbf{s}_t^{i-1}[l] - \mathbf{s}_t^{i-1}[r]) & , \text{if } l \neq r \\ -\sum_{j \neq r} \exp(\mathbf{s}_t^{i-1}[j] - \mathbf{s}_t^{i-1}[r]) & , \text{if } l = r \end{cases} \tag{37}$$

$$f_t^{i'}(\alpha) = \begin{cases} \exp(\mathbf{s}_t^{i-1}[l_t^i] + \alpha - \mathbf{s}_t^{i-1}[y_t]) & , \text{if } l_t^i \neq y_t \\ -\sum_{j \neq y_t} \exp(\mathbf{s}_t^{i-1}[j] - \alpha - \mathbf{s}_t^{i-1}[y_t]) & , \text{if } l_t^i = y_t. \end{cases} \tag{38}$$

With this gradient, if we set the learning rate $\eta_t^i = \frac{2\sqrt{2}}{(k-1)\sqrt{t}}e^{-i}$, a standard analysis provides $R^i(T) \leq 4\sqrt{2}(k-1)e^i\sqrt{T}$. Note that with exponential loss, we have different learning rate for each weak learner. We keep the algorithm same as Algorithm 2, but with different cost matrix and learning rate. Now we state the theorem for the mistake bound.

**Theorem 12. (Mistake bound with exponential loss)** *For any $T$ and $N$, the number of mistakes made by Algorithm 2 with above cost matrix and learning rate satisfies the following inequality with high probability:*

$$\sum_t \mathbb{1}(y_t \neq \hat{y}_t) \leq \frac{4k}{\sum_i \gamma_i^2}T + \tilde{O}\left(\frac{ke^{2N}}{\sum_i \gamma_i^2}\right).$$

*Proof.* The proof is almost identical to that of Theorem 5, and we only state the different steps. With cost matrix defined in (37), we can show

$$-\sum_t \mathbf{C}_t^i[y_t, y_t] \geq M_{i-1}.$$

Furthermore, we have following identity (which was inequality in the original proof):

$$L^{y_t}(\mathbf{s}_t^{i-1} + \alpha \mathbf{e}_{l_t^i}) - L^{y_t}(\mathbf{s}_t^{i-1}) = \begin{cases} \mathbf{C}_t^i[y_t, l_t^i](e^\alpha - 1) & \text{,if } l_t^i \neq y_t \\ \mathbf{C}_t^i[y_t, l_t^i](-e^{-\alpha} + 1) & \text{,if } l_t^i = y_t \end{cases}.$$

This leads to

$$\Delta_i \leq -\frac{\gamma_i^2}{2} M_{i-1} + 4\sqrt{2}(k-1)e^i\sqrt{T}.$$

Summing over $i$, we get

$$\frac{\sum_i \gamma_i^2}{2} \min_i M_i \leq (k-1)T + 4\sqrt{2}(k-1)e\frac{e^N - 1}{e - 1}\sqrt{T}$$

$$\leq (k-1)T + 9ke^N\sqrt{T}.$$

Plugging this in (32), we get with high probability,

$$\sum_t \mathbb{1}(y_t \neq \hat{y}_t) \leq \frac{4(k-1)}{\sum_i \gamma_i^2}T + \tilde{O}(\frac{ke^N\sqrt{T}}{\sum_i \gamma_i^2} + \log N)$$

$$\leq \frac{4k}{\sum_i \gamma_i^2}T + \tilde{O}(\frac{ke^{2N}}{\sum_i \gamma_i^2}),$$

which completes the proof. We also used AM-GM inequality for the last step. $\square$

Comparing to Theorem 5, we get a better coefficient for the first term, which is asymptotic error rate, but the exponential function in the second term makes the bound significantly loose. The exponential term comes from the larger variability of $f_t^i$ associated with exponential loss. It should also be noted that the empirical edge $\gamma_i$ is measured with different cost matrices, and thus direct comparison is not fair. In fact, as discussed in Section 4.1, $\gamma_i$ is closer to $0$ with exponential loss than with logistic loss due to larger variation in weights, which is another huge advantage of logistic loss.

## D.2   Square hinge loss

Another popular surrogate loss is square hinge loss. We begin the section by introducing multiclass version of it:

$$L^r(\mathbf{s}) := \frac{1}{2}\sum_{l \neq r}(\mathbf{s}[l] - \mathbf{s}[r] + 1)_+^2, \tag{39}$$

where $f_+ := \max\{0, f\}$. From this, we can compute the cost matrix and $f_t^{i'}$ for the online gradient descent as below:

$$\mathbf{C}_t^i[r, l] = \begin{cases} (\mathbf{s}_t^{i-1}[l] - \mathbf{s}_t^{i-1}[r] + 1)_+ & \text{,if } l \neq r \\ -\sum_{j \neq r}(\mathbf{s}_t^{i-1}[j] - \mathbf{s}_t^{i-1}[r] + 1)_+ & \text{,if } l = r \end{cases} \tag{40}$$

$$f_t^{i'}(\alpha) = \begin{cases} (\mathbf{s}_t^{i-1}[l_t^i] + \alpha - \mathbf{s}_t^{i-1}[y_t] + 1)_+ & \text{,if } l_t^i \neq y_t \\ -\sum_{j \neq y_t}(\mathbf{s}_t^{i-1}[j] - \alpha - \mathbf{s}_t^{i-1}[y_t] + 1)_+ & \text{,if } l_t^i = y_t \end{cases}. \tag{41}$$

With square hinge loss, we do not use Lemma 11 in the proof of mistake bound, and thus the feasible set $F$ can be narrower. In fact, we will set $F = [-c, c]$, where the parameter $c$ will be optimized later. With this $F$, we have $|f_t^{i'}(\alpha)| \leq (k-1) + ci \leq (k-1) + cN$, and the standard analysis of online gradient descent algorithm with learning rate $\eta_t = \frac{\sqrt{2}c}{((k-1)+cN)\sqrt{t}}$ provides that $R^i(T) \leq 2\sqrt{2}(k-1+cN)\sqrt{T}$. Now we are ready to prove the mistake bound.

**Theorem 13. (Mistake bound with square hinge loss)** *For any $T$ and $N$, with the choice of $c = \frac{1}{\sqrt{N}}$, the number of mistakes made by Algorithm 2 with above cost matrix and learning rate satisfies the following inequality with high probability:*

$$\sum_t \mathbb{1}(y_t \neq \hat{y}_t) \leq \frac{2k\sqrt{N}}{\sum_i |\gamma_i|}T + \tilde{O}(\frac{(k^2 + N)N\sqrt{N}}{\sum_i |\gamma_i|}).$$

*Proof.* With cost matrix defined in (40), we can show

$$-\sum_t \mathbf{C}_t^i[y_t, y_t] \geq M_{i-1}.$$

We can also check that

$$\frac{1}{2}[(s+\alpha)_+^2 - s_+^2] \leq s_+\alpha + \frac{\alpha^2}{2},$$

by splitting the cases with the sign of each term. Using this, we can deduce that

$$L^{y_t}(\mathbf{s}_t^{i-1} + \alpha\mathbf{e}_{l_t^i}) - L^{y_t}(\mathbf{s}_t^{i-1}) \leq \mathbf{C}_t^i[y_t, l_t^i]\alpha + \frac{(k-1)\alpha^2}{2}.$$

Summing over $t$ gives

$$\sum_t L^{y_t}(\mathbf{s}_t^{i-1} + \alpha\mathbf{e}_{l_t^i}) - L^{y_t}(\mathbf{s}_t^{i-1}) \leq \sum_t \mathbf{C}_t^i[y_t, y_t]\gamma_i\alpha + \frac{(k-1)\alpha^2}{2}T.$$

The RHS is a quadratic in $\alpha$, and the minimizer is $\alpha^* = -\frac{\sum_t \mathbf{C}_t^i[y_t, y_t]\gamma_i}{(k-1)T}$. Since the magnitude of $\mathbf{C}_t^i[y_t, y_t]$ grows as a function of $c$, there is no guarantee that this minimizer lies in the feasible set $F = [-c, c]$. Instead, we will bound the minimum by plugging in $\alpha = \pm c$:

$$\min_{\alpha \in [-c,c]} \sum_t L^{y_t}(\mathbf{s}_t^{i-1} + \alpha\mathbf{e}_{l_t^i}) - L^{y_t}(\mathbf{s}_t^{i-1}) \leq \frac{(k-1)c^2}{2}T + c|\gamma_i|\sum_t \mathbf{C}_t^i[y_t, y_t]$$

$$\leq \frac{(k-1)c^2}{2}T - c|\gamma_i|M_{i-1}.$$

From this, we get

$$\Delta_i \leq -c|\gamma_i|M_{i-1} + \frac{(k-1)c^2}{2}T + 2\sqrt{2}(k-1+cN)\sqrt{T}.$$

Summing over $i$, we get

$$c\sum_i |\gamma_i| \min_i M_i \leq \frac{k-1}{2}T + \frac{(k-1)c^2 N}{2}T + 2\sqrt{2}(k-1+cN)N\sqrt{T}.$$

By rearranging terms, we conclude

$$\min_i M_i \leq \frac{(k-1)}{2\sum_i |\gamma_i|}\left(\frac{1}{c} + cN\right)T + \frac{2\sqrt{2}(k-1+cN)N}{\sum_i |\gamma_i|}\sqrt{T}.$$

It is the first term from the RHS that provides an optimal choice of $c = \frac{1}{\sqrt{N}}$, and this value gives

$$\min_i M_i \leq \frac{(k-1)\sqrt{N}}{\sum_i |\gamma_i|}T + \frac{2\sqrt{2}(k-1+\sqrt{N})N}{\sum_i |\gamma_i|}\sqrt{T}.$$

Plugging this in (32), we get with high probability,

$$\sum_t \mathbb{1}(y_t \neq \hat{y}_t) \leq \frac{2(k-1)\sqrt{N}}{\sum_i |\gamma_i|}T + \tilde{O}\left(\frac{(k+\sqrt{N})N}{\sum_i |\gamma_i|}\sqrt{T} + \log N\right)$$

$$\leq \frac{2k\sqrt{N}}{\sum_i |\gamma_i|}T + \tilde{O}\left(\frac{(k^2+N)N\sqrt{N}}{\sum_i |\gamma_i|}\right),$$

which completes the proof. We also used AM-GM inequality for the last step. □

By Cauchy-Schwartz inequality, we have $N\sum_i \gamma_i^2 \geq (\sum_i |\gamma_i|)^2$. From this, we can deduce $(\frac{\sqrt{N}}{\sum_i |\gamma_i|})^2 \geq \frac{1}{\sum_i \gamma_i^2}$. If LHS is greater than 1, then the bound in Theorem 13 is meaningless. Otherwise, we have

$$\frac{\sqrt{N}}{\sum_i |\gamma_i|} \geq \left(\frac{\sqrt{N}}{\sum_i |\gamma_i|}\right)^2 \geq \frac{1}{\sum_i \gamma_i^2},$$

which validates that the bound with logistic loss is tighter. Furthermore, square hinge loss also produces more variable weights over instances, which results in worse empirical edges.

# Appendix E Detailed description of experiment

Testing was performed on a variety of data sets described in Table 2. All are from the UCI data repository (Blake and Merz [22], Higuera C [23], Ugulino et al. [24]) with a few adjustments made to deal with missing data and high dimensionality. These changes are noted in the table below. Many of the data sets are the same as used in the Oza [5], with the addition of a few sets with larger numbers of data points and predictors. We report the average performance on both the entire data set and on the final 20% of the data set. The two accuracy measures help understand both the "burn in period", or how quickly the algorithm improves as observations are recorded, and the "accuracy plateau", or how well the algorithm can perform given sufficient data. Different applications may emphasize each of these two algorithmic characteristics, so we choose to provide both to the reader. We also report average run times. All computations were carried out on a Nehalem architecture 10-core 2.27 GHz Intel Xeon E7-4860 processors with 25 GB RAM per core. For all but the last two data sets, results are averaged over 27 reordering of the data. Due to computational constraints, Movement was run just nine times and ISOLET just once.

Table 2: Data set details

| Data sets | Number of data points | Number of predictors | Number of classes |
|---|---|---|---|
| Balance | 625 | 4 | 3 |
| Mice | 1080 | 82* | 8 |
| Cars | 1728 | 6 | 4 |
| Mushroom | 8124 | 22 | 2 |
| Nursery | 12960 | 8 | 4 |
| ISOLET | 7797 | 50** | 26 |
| Movement | 165631*** | 12*** | 5 |

* Missing data was replaced with 0.
** The original 617 predictors were projected onto their first 50 principal components, which contained 80% of the variation.
*** User information was removed, leaving only sensor position predictors. Single data point with missing value removed.

In all the experiments we used Very Fast Decision Trees (VFDT) from Domingos and Hulten [14] as weak learners. VFDT has several tuning parameters which relate to the frequency with which the tree splits. In all methods we assigned these randomly for each tree. Specifically for our implementation the tuning parameter `grace_period` was chosen randomly between 5 and 20 and the tuning parameters `split_confidence` and `hoeffding_tie_threshold` randomly between 0.01 and 0.9. It is likely that this procedure would produce trees which do not perform well on specific data sets. In practice for the Adaboost.OLM it is possible to restart poorly performing trees using parameters similar to better performing trees in an automated and online (although ad hoc) fashion using the $\alpha_t^i$, and this tends to produce superior performance (as well as allow adaptivity to changes in the data distribution). However for these experiments, we did not take advantage of this to better examine the benefits of just the cost matrix framework.

Several algorithms were tested using the above specifications, but with slightly different conditions. The first three are directly comparable since they all use the same weak learners and do not require knowledge of the edge of the weak learners. DT is the best result from running 100 VFDT independently. The best was chosen after seeing the performance on the entire data set and final 20% respectively. However the time reported was the average time for running all 100 VFDT. This was done to better see the additional cost of running the boosting framework on top of the training of the raw weak learners. OLB is an implementation of the Online Boosting algorithm in Oza [5, Figure 2] with 100 VFDT. AdaOLM stands for Adaboost.OLM, again with 100 VFDT.

The next five algorithms (MB) tested were all variants of the OnlineMBBM but with different edge $\gamma$ values. In practice this value is never known ahead of time, but we want to explore how different edges affect the performance of the algorithm. For the ease of computation, instead of exactly finding the value of (16), we estimated the potential functions by Monte Carlo (MC) simulations.

Table 3: Comparison of algorithms on final 20% of data set

| Data sets | 100 multiclass trees | | | | | | | | 100$k$ binary trees | |
|---|---|---|---|---|---|---|---|---|---|---|
| | DT | OLB | AdaOLM | MB .3 | MB .1 | MB .05 | MB .01 | MB .001 | OvA | AdaOVA |
| Balance | 0.768 | 0.772 | 0.754 | 0.788 | 0.821 | 0.819 | 0.805 | 0.752 | 0.786 | 0.795 |
| Mice | 0.608 | 0.399 | 0.561 | 0.572 | 0.695 | 0.663 | 0.502 | 0.467 | 0.742 | 0.667 |
| Cars | 0.924 | 0.914 | 0.930 | 0.914 | 0.885 | 0.870 | 0.836 | 0.830 | 0.946 | 0.919 |
| Mushroom | 0.999 | 1.000 | 1.000 | 0.997 | 1.000 | 1.000 | 0.999 | 0.998 | 1.000 | 1.000 |
| Nursery | 0.953 | 0.941 | 0.966 | 0.965 | 0.969 | 0.964 | 0.948 | 0.940 | 0.974 | 0.965 |
| ISOLET | 0.515 | 0.149 | 0.521 | 0.453 | 0.626 | 0.635 | 0.226 | 0.165 | 0.579 | 0.570 |
| Movement | 0.915 | 0.870 | 0.962 | 0.975 | 0.987 | 0.988 | 0.984 | 0.981 | 0.947 | 0.970 |

Table 4: Comparison of algorithms on full data set

| Data sets | 100 multiclass trees | | | | | | | | 100$k$ binary trees | |
|---|---|---|---|---|---|---|---|---|---|---|
| | DT | OLB | AdaOLM | MB .3 | MB .1 | MB .05 | MB .01 | MB .001 | OvA | AdaOVA |
| Balance | 0.734 | 0.747 | 0.698 | 0.751 | 0.769 | 0.759 | 0.736 | 0.677 | 0.724 | 0.730 |
| Mice | 0.499 | 0.315 | 0.454 | 0.457 | 0.507 | 0.449 | 0.356 | 0.343 | 0.586 | 0.530 |
| Cars | 0.848 | 0.839 | 0.865 | 0.842 | 0.829 | 0.814 | 0.767 | 0.762 | 0.881 | 0.853 |
| Mushroom | 0.996 | 0.997 | 0.995 | 0.991 | 0.995 | 0.994 | 0.993 | 0.992 | 0.996 | 0.995 |
| Nursery | 0.921 | 0.909 | 0.928 | 0.932 | 0.936 | 0.932 | 0.918 | 0.912 | 0.939 | 0.932 |
| ISOLET | 0.395 | 0.104 | 0.456 | 0.333 | 0.486 | 0.461 | 0.152 | 0.111 | 0.507 | 0.472 |
| Movement | 0.898 | 0.864 | 0.942 | 0.954 | 0.972 | 0.973 | 0.959 | 0.957 | 0.927 | 0.952 |

Table 5: Comparison of algorithms total run time in seconds

| Data sets | 100 multiclass trees | | | | | | | | 100$k$ binary trees | |
|---|---|---|---|---|---|---|---|---|---|---|
| | DT | OLB | AdaOLM | MB .3 | MB .1 | MB .05 | MB .01 | MB .001 | OvA | AdaOVA |
| Balance | 8 | 19 | 20 | 26 | 42 | 47 | 50 | 51 | 66 | 43 |
| Mice | 105 | 263 | 416 | 783 | 2173 | 3539 | 3579 | 3310 | 3092 | 3013 |
| Cars | 39 | 27 | 59 | 56 | 105 | 146 | 165 | 152 | 195 | 143 |
| Mushroom | 241 | 169 | 355 | 318 | 325 | 326 | 324 | 321 | 718 | 519 |
| Nursery | 526 | 302 | 735 | 840 | 1510 | 2028 | 2181 | 1984 | 2995 | 1732 |
| ISOLET | 470 | 1497 | 2422 | 18732 | 38907 | 64707 | 62492 | 50700 | 37300 | 33328 |
| Movement | 1960 | 3437 | 5072 | 13018 | 17608 | 18676 | 16739 | 16023 | 30080 | 21389 |

The final two algorithms are slightly different implementations of the One VS All (OvA) ensemble method. In this framework multiple binary classifiers are used to solve a multiclass problem by viewing different classes as the positive class, and all others as the negative class. They then predict whether a data point is their positive class or not, and the results are used together to make a final classification. Both use VFDT as their weak learners, but with $100 \times k$ binary trees. The first method (OvA) uses $k$ versions of Adaboost.OL, each viewing one of the classes as the positive class. Recall that Adaboost.OLM in the binary setting is just Adaboost.OL by Beygelzimer et al. [7]. The second (AdaOVA) produces 100 weak multiclass classifiers by grouping a $k$ binary classifiers, one for each class, and then uses Adaboost.OLM to get the final learner, treating the 100 single tree OvA's as its weak learners. In the table below we have partitioned the methods in terms of the number of weak learners since, while they all tackle the same problem, algorithms within each partition are more directly comparable since they use the same weak learners.

## E.1 Analysis

It is worth beginning by noting the strength of the VFDT without any boosting framework. While the results above are for the best performing tree in hindsight, which is not a valid strategy in practice, in many applications it would be possible to collect some data beforehand activating the system, and

use that to pick tuning parameters. It is also worth noting that many of the weaknesses of the above methods, such as their poor scaling with the number of predictors, are also inherited from the VFDT. Nonetheless in almost all cases Adaboost.OLM algorithm outperforms both the best tree and the preexisting Online Boosting algorithm (and is often comparable to the OnlineMBBM algorithms), as well as provide theoretical guarantees. In particular these performance gains seem to be greater on the final 20% of the data and in data sets with larger number of data points $n$, leading us to believe that Adaboost.OLM has a longer burn in period, but higher accuracy plateau. This performance does come at additional computational cost, but this cost is relatively mild, especially compared to the costs of OnlineMBBM and the OvA methods.

The OnlineMBBM methods use additional assumptions about the power of their weak learners, and are able to leverage that additional information to produce more accurate, with one of these algorithms often achieving the highest accuracy on each data set. However they can be sensitive to the choice of $\gamma$, with the worst choice of $\gamma$ often underperforming both pure trees and Adaboost.OLM, and with no single $\gamma$ value always producing the best result. These methods are also much slower than Adaboost.OLM, likely due to computational burden in estimating the potential functions.

Finally our two OvA algorithms tend to perform very well, often beating the other adaptive methods. However this performance is likely due to the use of many times more weak learners than the other adaptive methods used, which results in high computational cost. Again we see that as $n$ increases the implementation of OvA using our cost matrix framework performs better compared to the vanilla implementation, reinforcing our belief that the cost matrix framework requires more data to come online but has a higher accuracy plateau.