[Reviews · NeurIPS 2017]

Reviewer 1



This paper aims to develop online version of the multiclass boosting algorithm. The authors combine the ideas in [4] and [7] I order to develop their framework. Overall, the paper is well written and easy to follow. It also seems that framework is technically sound. However, there are certain parts of the paper that is not well-justified or motivated. I talk about them below: 1- The title of Section 3 is “an optimal algorithm”. It is not clear in what sense the developed algorithm in this section is optimal. There is no theory proving that the bounds obtained using the developed algorithm is optimal for online multi-class boosting. If mistakes bounds of O(T) (for example in case of 0-1 loss) is optimal for online multi-class boosting, it should be proved or there should be some reference to where is has already been shown. 2- Now, let's say it can be proved that such bounds are optimal for online boosting. What is the motivation behind online boosting when the bounds are suboptimal when compared to online learning with expert advice (O(log(T))? Why having multiple weak learners are useful at all when it doesn’t lead to better results in terms of (mistake) bounds compared to standard online learning models? 3- Given that in online boosting, you already have all the weak learners available, why do you still calculate the potential function sequentially, assuming you don’t have the knowledge of labels for any weak learner after the current weak learner i. This is reflected in designing the potential function in Equation (3) which results in computing the weights w^i(t) and cost matrix D_t^i sequentially. To me, it seems that computing the potential function by using the outputs of all weak learners is more precise and I am not sure why such an approach is not taken here. Will that result in better bounds? And here are some minor problems: Section 3.1: what do you mean by “For optimal algorithm, we assume that \alpha_t^{i}=1, \forall i,t”? The notation is sometimes difficult. As an example, it took me some times to understand s(r) in Section 3.1 refers to element r of vector s. I would have used s[r] instead of s(t) to prevent the confusion with a function. Section 3.2: “Under 0-1 loss” is repeated twice in 3rd sentence.

Reviewer 2



This paper develops appropriate theory and proposes two algorithms for online multi class boosting. One particularly notable result is a weak learning condition for online multi class boosting. This paper would be a notable addition to this recently growing literature, therefore, I recommend acceptance. A few comments: 1. On page 3, in definition 1, it is worth commenting on whether there are any theoretical developments describing how S, the excess loss, decreases with the number of training examples. This would seem a basic requirement for weak learning in an online setting. 2. In the experimental section, it is worth noting that the comparison to the original online boosting algorithm is not quite fair since it was developed as a two-class algorithm. The experimental results on the online boosting algorithm are worth retaining because, as the authors point out, it is a good baseline.

Reviewer 3



In this paper the authors introduce two learning algorithms dealing with online learning in a multi-class setting. The novelty resides in combining a multi-class boosting framework providing solid theoretical results with an online learning framework. Notable contributions of the paper include: a weak learning condition for multi-class online learning, a general boosting algorithm and a more practical version. I have two main concerns about this paper. The paper is overall well written, the key ideas are ordered in an easy to follow manner and, as far as I can tell, the theoretical results are sound. However, at times the paper is hard to follow without switching over to the supplementary material. For instance, the link between the weak learning condition in batch learning and the one presented in the paper, is not obvious without reading the supplementary material; Section 3.2 is also hard to follow since the supplementary material is needed in order to grasp key ideas. While I do understand the authors' eagerness to include most of their work in the paper, in my opinion, the main paper should be as close as possible a stand alone paper, which isn't the case for this paper. My second concern is related to the experimental and (the missing) discussion sections. The experimental results suggest that the proposed methods can achieve better results than other online methods, however their run time is obviously far more important. As such two questions arise : 1) Is there a case where the proposed methods are more interesting to use than existing ones wrt the ratio performance/run time? 2) Is it possible to speed up the proposed methods without significant loss in their performances? A discussion section treating both questions would've been appreciated. I also think that more experimental results are needed in order to evaluate how the proposed methods scale wrt the number of classes and examples. When introducing the adaptive algorithm the authors propose to discard the weak learning condition and introduce an empirical edge. However the empirical edge takes its values in [-1,1], while the edge in the weak learning condition takes its values in (0,1). In practice, how is a negative edge dealt with? Is the training sample discarded?